# GAMA: A Neural Neighborhood Search Method with Graph-aware Multi-modal Attention for Vehicle Routing Problem

## Abstract

Recent advances in neural neighborhood search methods have shown potential in tackling Vehicle Routing Problems (VRPs). However, most existing approaches rely on simplistic state representations and fuse heterogeneous information via naive concatenation, limiting their ability to capture rich structural and semantic context. To address these limitations, we propose GAMA, a neural neighborhood search method with Graph-aware Multi-modal Attention model in VRP. GAMA encodes the problem instance and its evolving solution as distinct modalities using graph neural networks, and models their intra- and inter-modal interactions through stacked self- and cross-attention layers. A gated fusion mechanism further integrates the multi-modal representations into a structured state, enabling the policy to make informed and generalizable operator selection decisions. Extensive experiments conducted across various synthetic and benchmark instances demonstrate that the proposed algorithm GAMA significantly outperforms the recent neural baselines. Further ablation studies confirm that both the multi-modal attention mechanism and the gated fusion design play a key role in achieving the observed performance gains.

## 1 Introduction

In recent years, Learning to Optimize (L2O) Kool et al. (2018); Joshi et al. (2019); Hottung et al. (2021) has emerged as a promising paradigm for solving combinatorial problems like VRP by training data-driven models to learn optimization strategies from experience. Unlike traditional hand-designed heuristics, L2O approaches can adapt to new problem instances, generalize across distributions, and leverage structural patterns in data, making them an attractive alternative for scalable and automatic optimization. Within the L2O framework, a growing body of work focuses on Learning to Improve (L2I) Wang et al. (2024); Kong et al. (2024); Sultana et al. (2024) methods, which iteratively refine a given (possibly suboptimal) solution through the application of predefined local search operators. Compared to end-to-end construction policies (L2C) Bi et al. (2024); Lin et al. (2024); Mozhdehi et al. (2024); Liu et al. (2025), this approach naturally aligns with the VRP search process, where the solution quality is typically improved through iterative local modifications. By mimicking this improvement-based strategy, L2I enables the agent to effectively navigate the solution space and escape from poor local optima.

In this work, we focus on the L2I framework with operator selection, a type of neural neighborhood search method for VRP, where each policy decision involves selecting the most suitable operator to apply to the current solution. This formulation treats the operator as the atomic action in a reinforcement learning framework, and the policy is trained to select operators that maximize long-term solution quality. While this approach holds strong potential, its effectiveness hinges on two crucial components: the quality of the learned state representation and the capability of the policy to make informed operator selection decisions.

However, most existing neural neighborhood search methods rely on simplistic or coarse-grained features extracted from high-level signals such as objective values of the current solution Choong et al. (2018), the last applied operator Qi et al. (2022), or static instance descriptors Yi et al. (2022). These representations often fail to capture the structural and spatial characteristics embedded

within the evolving solution, which are critical for accurately understanding the current search state. Furthermore, although some studies attempt to incorporate diverse features (e.g., historical trajectories, instance characteristics, and solution metrics), these heterogeneous inputs are typically combined via simple concatenation Lu et al. (2019); Guo et al. (2025). Such an approach fails to capture the underlying semantic relationships among the inputs, potentially causing representational entanglement and scale inconsistency. These limitations can degrade the learned policy's ability to generalize across different instances and search scenarios.

To overcome the limitations mentioned above, we propose GAMA, a novel **g**raph-**a**ware **m**ultimodal **a**ttention model for neural neighborhood search in VRP. GAMA captures semantic interactions between the problem instance and its current solution through structured GNN encoding and attention-based fusion. The learned representation offers informative context to guide the selection of effective neighborhood operators. Our main contributions are summarized as follows:

1. We present an effective neural neighborhood search method for VRP that adaptively selects search operators based on the current search state, enabling dynamic and informed solution improvement.

2. We design a graph-aware multimodal attention encoder that independently encodes the VRP instance and current solution as distinct semantic modalities using graph convolutional networks. Intra-modality and inter-modality dependencies are modeled through stacked self-attention and cross-attention mechanisms.

3. We incorporate a gated fusion module to integrate the multimodal representations, providing the policy network with rich, structured state features that improve generalization and decision-making quality across diverse problem instances.

## 2    RELATED WORK

Recently, learning-based approaches have emerged as a promising alternative by enabling data-driven, adaptive decision-making. Within the Learning-to-Optimize (L2O) paradigm, three main sub-fields have gained prominence: learning-to-construct (L2C) Luo et al. (2025), learning-to-predict (L2P) solvers, and learning-to-improve (L2I) Ma et al. (2022); Hottung et al. (2025); Ma et al. (2023); Ouyang et al. (2025). Related literature can be found in the appendix A.1.

Adaptive Operator Selection (AOS) aims to automatically choose the most suitable operator at each decision point to guide the search process effectively. With the rise of machine learning, particularly reinforcement learning (RL) Guo et al. (2025); Liao et al. (2025), learning-based AOS Pei et al. (2025) has emerged as a promising direction that formulates operator selection as a sequential decision-making problem Aydin et al. (2024). This paradigm aligns with the broader Learning to Improve (L2I) framework, where learning agents are trained to iteratively refine solutions.

Despite its potential, learning-based AOS faces several critical challenges:

(1) How to construct informative state representations:

A key to effective operator selection lies in how the state of the search process is represented. Most existing approaches use macro-level handcrafted features, such as objective values Lu et al. (2019), operator usage history Qi et al. (2022), solution diversity Handoko et al. (2014), and computational resources consumed/left Dantas & Pozo (2022). These features offer abstract insights but often fail to reflect the fine-grained structural details of the current solution. Especially for combinatorial problems like VRP or TSP, where the solution space is graph- or sequence-structured, such high-level features are insufficient. While macro features are relatively easy to design, micro-level features—such as solution structure, partial tours, or local neighborhoods—offer a more direct and fine-grained view of the ongoing search. For problems with fixed-length solution encodings (e.g., continuous optimization or knapsack), vector-based representation is effective Tian et al. (2022). However, in routing or scheduling problems, where solutions are inherently combinatorial and dynamic, micro-level representation is less explored due to its complexity. Although some efforts encode static problem structures using GNNs or attention mechanisms Duan et al. (2020); Lei et al. (2022), they typically overlook how solutions evolve and how operators transform them, which limits their effectiveness in adaptive decision-making.

(2) How to integrate heterogeneous information sources effectively:

Some Learning-based methods attempt to incorporate diverse types of input features—such as solution embeddings, operator identity, historical usage, and search trajectory. A common practice is direct feature concatenation Guo et al. (2025); Lu et al. (2019), but this approach ignores the semantic heterogeneity among these inputs. Such naive integration may result in feature redundancy or conflict, thereby degrading the quality of learned policies. A principled fusion mechanism is needed to resolve semantic inconsistencies and encourage synergy across modalities.

As a result, while existing learning-based AOS frameworks have demonstrated promising results, they still face limitations in state encoding granularity, operator-context modeling, and semantic feature fusion.

## 3 Methodology

We now present the details of our Graph-Aware Multimodal Attention model (GAMA), designed to enable adaptive neural neighborhood search for VRP through structured learning and RL-based operator selection.

### 3.1 Overall Framework

The proposed GAMA framework is built upon a local search-based optimization process, aiming to adaptively select operators during the search, such as 2-opt, swap, insertion and so on. Unlike traditional methods that rely on fixed or handcrafted operator sequences, GAMA leverages structural representations of both the problem instance, evolving solution, and optimization history to guide operator selection dynamically. The details of the operators are presented in supplementary material. Once an operator is selected, it is applied exhaustively in the neighborhood of the current solution, the best improving move is then adopted to update the solution. Figure 1 illustrates the overall architecture of GAMA, and Algorithm 1 summarizes the procedural flow.

For each episode $m$, the $m$-th problem instance is loaded, and the initial solution $\delta$ is constructed (line 4). Given the current policy, the solution is then iteratively refined over $T$ steps. At each iteration step $t$, the current state $s_t$, together with the corresponding problem instance, is encoded by the GAMA encoder into a unified representation $s_t$. Given this representation, the RL agent parameterized by $\theta$ selects an operator $a_t \sim \pi(\cdot|s_t; \theta)$ from a set of low-level local search operators. The selected operator $a_t$ is applied to transform the current solution into a new solution $\delta_{t+1}$. This transition $\langle \delta_t, a_t, \delta_{t+1} \rangle$ is stored in an experience memory buffer $\mathcal{B}$, which is then used to update the policy network after $T$ steps. To escape local optima, GAMA monitors the progress of the best-found solution. If no improvement is observed for $L$ consecutive iterations, a shake procedure Mladenović & Hansen (1997) is triggered to perturb the current solution using a randomly selected operator, enhancing long-term exploration (lines 15-16). The iteration continues until a termination condition is met, such as reaching the maximum number of steps $T$. This process is repeated across multiple problem instances for $NoE$ episodes. Upon completion, the policy $\pi_\theta(\cdot)$ is trained and ready for deployment.

### 3.2 Markov Decision Process (MDP)

The agents' selection procedure can be modeled as a Markov Decision Process (MDP), where the action space consists of operator choices, and the environment transitions are defined by applying the best move in the selected operator's neighborhood:

**State.** At time $t$, the state is defined to include 1) problem features, 2) features of the current solution, and 3) optimization history, i.e.,

$$s_t = \{\mathcal{G}_{\text{dis}}, \mathcal{G}_{\text{sol}}, \mathcal{X}_t, a, e, \Delta, \eta\} \tag{1}$$

where $\mathcal{G}_{\text{dis}}$ denotes the distance graph, whose edge weights represent the Euclidean distance between customer nodes; $\mathcal{G}_{\text{sol}}$ denotes the solution graph, indicating the current solution topology; $\mathcal{X}_t$ represents the node features at time $t$; the full definition of $\mathcal{G}_{\text{dis}}$, $\mathcal{G}_{\text{sol}}$, and $\mathcal{X}_t$ is deferred to the supplementary material; $a$ denotes the operator (action) selected at the previous step; $e \in \{-1, 1\}$ is a binary indicator representing whether the previous action $a$ was effective (i.e., whether it led

**Algorithm 1:** GAMA Learning Process

---

**Input**   : Maximum episodes $NoE$, maximum timesteps $T$, max no improvement threshold $L$
**Output** : the learned policy $\pi_\theta(\cdot)$.
1 Randomly initialize the policy $\pi_\theta(\cdot)$
2 **for** *episode* $m = 1$ *to* $NoE$ **do**
3      // Initialization
4      Load instance and construct initial solution $\delta$
5      Initialize experience memory buffer $\mathcal{B} \leftarrow \emptyset$
6      // Iterative local search process
7      **for** *timestep* $t = 1$ *to* $T$ **do**
8          $k = 0$; Extract the state feature and set state $s_t$ from GAMA encoder.
9          Select the next operator: $a_t \leftarrow \pi_\theta(s_t))$
10         Apply operator and update solution: $\delta_{t+1} \leftarrow$ **Local Search**$(\delta_t, a_t)$
11         Save experience $\langle \delta_t, a_t, \delta_{t+1} \rangle$ to $\mathcal{B}$
12         **if** $f(\delta_{t+1}) < f(\delta^*)$ **then**
13            Update $\delta^* = \delta_t$    $C_{notI} \leftarrow 0$
14         **else**
15            $C_{notI} \leftarrow C_{notI} + 1$
16         $t = t + 1$
17         **if** $C_{notI} \geq L$ **then**
18            $k = k + 1$
19            Compute the phase reward: $r^{(k)} = f(\delta^{(0)}) - f(\delta_{(k)}^*)$
20            Assign $r^{(k)}$ to all transitions of this phase in $\mathcal{B}$
21            Apply shake: $\delta_t \leftarrow$ **Shake** $(\delta_t)$
22      // Policy learn and update
23      Sample random mini-batch of experiences from $\mathcal{B}$ and Update $\pi_\theta$ using mini-batch.
24 **END**

---

to an improved solution). $\Delta$ is the gap between the current solution and the current best solution, i.e., $\Delta = f(\delta) - f(\delta^*)$, where $\delta$ is the current solution and $\delta^*$ is the current best solution so far; $\eta$ measures the change in objective value caused by the last action, defined as the difference between the current solution cost and that of the previous step.

**Action.** The action $a \in \mathcal{A}$ refers to the selection of a specific local search operator at iteration step $t$, where $\mathcal{A}$ is the predefined operator set.

**Reward.** Let a single improvement phase $k$ be defined as the sequence of operator applications between two consecutive shake operations. The reward function is defined as $r_t = f(\delta_0) - f(\delta_{(k)}^*), \forall t \in \mathcal{T}_k$, which is computed at the end of each improvement phase $k$. $\delta^{(0)}$ is the initial solution of the phase and $\delta_{(k)}$ is the best solution obtained within this phase. All operators used in the same iteration will receive the same reward Lu et al. (2019), calculated as the cost difference between the initial and current best solution found during this improvement phase $k$.

**Policy.** The policy $\pi_\theta$ governs the selection of local search operators based on the current state $s_t$, which is parameterized by the proposed GAMA model with parameters $\theta$.

**State Transition.** The next state $s_{t+1}$ is originated from $s_t$ by performing the selected operator $a_t$ on the current solution, i.e., $\mathcal{P} : s_t \xleftarrow{a_t} s_{t+1}$, which is tied to the solution transformation. Specifically, the solution transformation under the search policy is defined as $\Delta_{\text{search}} : \delta \xrightarrow{a} \delta' =$ LocalSearch$(\delta)$, where $\delta'$ denotes the best neighbor obtained by exhaustively evaluating all candidate neighbors of the current solution $\delta$ within the defined neighborhood.

### 3.3 GAMA Encoder

In GAMA, the encoder plays a critical role in transforming the raw input—comprising the problem instance, the current solution, and the search dynamics—into a compact, informative representation that guides the reinforcement learning (RL) agent.

Unlike traditional encoders that focus solely on static problem structure or solution states, our encoder is designed to integrate three complementary sources of information (as illustrated in Fig-

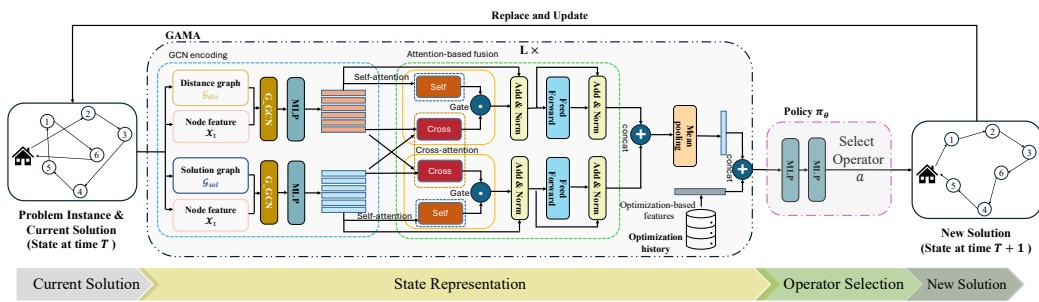

Figure 1: Illustration of iteration step within the proposed GAMA method.

ure 1) : 1) the problem instance graph, 2) the solution graph, and 3) the optimization trajectory features. To achieve this, the GAMA encoder is composed of a Dual-GCN module followed by $L = 3$ stacked attention-based fusion layers. Specifically, the Dual-GCN independently processes the instance-level topology and the dynamic solution-specific structure, generating two sets of node-level embeddings that reflect both the global problem layout and the local search status. These two modalities are then passed into the attention-based fusion encoder, which models both inter- and intra-graph interactions via multi-head cross-and self-attention, allowing the model to adaptively highlight salient structural and behavioral patterns. In parallel, we embed handcrafted optimization features into a compact global context vector that reflects the current progress of the search process. Finally, the fused graph features are concatenated with the optimization context vector to form the final state embedding. This embedding serves as the input to the policy network, enabling the RL agent to make informed, context-aware operator selections. By jointly capturing problem geometry, solution evolution, and search trajectory context, the resulting representation provides a rich and adaptive state encoding tailored for effective adaptive operator selection. Details are introduced as follows.

### 3.3.1 Dual-GCN Module

To simultaneously capture both the static structure of the VRP instance and the evolving dynamics of the current solution, we design a Dual-GCN module, which contains two separate graph convolutional encoders: one for the original problem instance graph $\mathcal{G}_{\text{dis}}$, and one for the current solution graph $\mathcal{G}_{\text{sol}}$. These two branches operate in parallel, encoding different but complementary aspects of the optimization state.

Given a shared input node feature matrix $X_t \in \mathbb{R}^{|V| \times d}$ at time step $t$ (including node coordinates, demands, vehicle load, etc.), we apply a standard graph convolutional network (GCN) to encode the structural information of the problem instance and the current solution separately:

$$H_{\text{dis}} = \sigma \left( \tilde{D}_{\text{dis}}^{-\frac{1}{2}} \tilde{\mathcal{G}}_{\text{dis}} \tilde{D}_{\text{dis}}^{-\frac{1}{2}} \mathcal{X}_t W_{\text{dis}} \right) \quad H_{\text{sol}} = \sigma \left( \tilde{D}_{\text{sol}}^{-\frac{1}{2}} \tilde{\mathcal{G}}_{\text{sol}} \tilde{D}_{\text{sol}}^{-\frac{1}{2}} \mathcal{X}_t W_{\text{sol}} \right) \tag{2}$$

Here, $\sigma(\cdot)$ stands for the activation function; $\tilde{\mathcal{G}} = \mathcal{G} + I$ is the adjacency matrix with added self-loops; $\tilde{D}$ is the corresponding diagonal degree matrix of $\tilde{\mathcal{G}}$; $W_{\text{dis}}, W_{\text{sol}}$ are learnable weight matrices for each GCN stream.

### 3.3.2 Attention-based Fusion Module

After obtaining the dual node-level representations $H_{\text{dis}}, H_{\text{sol}} \in \mathbb{R}^{|V| \times d_{\text{hid}}}$ from the Dual-GCN module, we employ a multi-layer attention-based fusion encoder to model both intra- and inter-modality interactions between the problem instance structure and the evolving solution status. This allows the encoder to dynamically align and refine the two information streams to form a unified and context-aware representation. At each fusion layer, we apply the following operations:

**(1) Self-Attention: Intra-Graph Encoding**   To capture the internal dependencies within each graph modality, we apply a multi-head self-attention mechanism to both the distance graph embeddings $H^{\text{dis}}$ and the solution graph embeddings $H^{\text{sol}}$.

Let the input embedding of a given modality at time step $t$ be $H \in \mathbb{R}^{|V| \times d}$, where $|V|$ is the number of nodes and $d$ is the embedding dimension. For each attention head $m \in \{1, \ldots, M\}$, we compute

$$Q_m = HW_m^Q, \quad K_m = HW_m^Q \quad V_m = HW_m^V, \tag{3}$$

where $W_m^Q, W_m^K, W_m^V \in \mathbb{R}^{d \times d_k}$ are learnable projection matrices and $d_k = d/M$ is the dimension per head. The attention output for head $m$ is

$$\text{head}_m = \text{softmax}\left(\frac{Q_m K_m^\top}{\sqrt{d_k}}\right) V_m, \quad \text{head}_m \in \mathbb{R}^{|V| \times d_k}. \tag{4}$$

The outputs of all $M$ heads are concatenated along the feature dimension and projected back to $d$:

$$H^s = \text{Concat}(\text{head}_1, \ldots, \text{head}_M)W^O, \quad W^O \in \mathbb{R}^{Md_k \times d}. \tag{5}$$

Thus, $H^s \in \mathbb{R}^{|V| \times d}$ preserves the original dimensionality.

This step allows the model to emphasize locally salient patterns such as customer clusters or over-congested sub-routes. We denote the outputs of two self-attention modules in Figure 1 as $H_{\text{dis}}^s$ and $H_{\text{sol}}^s$, respectively.

**(2) Cross-Attention: Inter-Graph Alignment**   The cross-attention module is designed to capture inter-modal interactions between problem features and current solution features. The core idea behind this module is to learn pairwise associations between the two modalities and then propagate information from one to the other accordingly. In the following part, we introduce the cross-attention mechanism in detail.

To model the associations between the problem and solution feature sequences, we first transform each modality into three components — query, key, and value — through learned linear projections. For convenience, we use the single-head attention mechanism to describe this process. Then, the outputs are calculated as:

$$H_{\text{dis}}^c = \text{softmax}\left(\frac{Q_{\text{dis}} K_{\text{sol}}^T}{\sqrt{d_k}}\right) V_{\text{sol}} \quad H_{\text{sol}}^c = \text{softmax}\left(\frac{Q_{\text{sol}} K_{\text{dis}}^T}{\sqrt{d_k}}\right) V_{\text{dis}} \tag{6}$$

These operations allow each node in the distance graph to attend to the solution structure and vice versa, learning how current routing decisions relate to the underlying problem geometry.

**(3) Gated Fusion: Adaptive Feature Integration**   To balance the retained modality-specific features and the cross-enhanced signals, we introduce a gating mechanism to adaptively fuse the self- and cross-attention outputs:

$$\tilde{H} = \alpha \odot H^s + (1 - \alpha) \odot H^c \quad \text{where} \quad \alpha = \sigma\big([H^s; H^c]W_g\big). \tag{7}$$

Here, $[H^s; H^c] \in \mathbb{R}^{|v| \times 2d}$ denotes the concatenation along the feature dimension; $W_g \in \mathbb{R}^{2d \times d}$ is a learnable projection, $\odot$ is element-wise multiplication; and $\sigma$ is the sigmoid function.

We denote the two outputs of gate layers in Figure 1 by $\tilde{H}_{\text{dis}}$ and $\tilde{H}_{\text{sol}}$, respectively. This gating unit enables the model to control how much cross-modal information should influence each node representation, mitigating potential negative interference from noisy alignment.

The resulting fused embeddings are first processed by residual connections He et al. (2016) and layer normalization (LN) Ba et al. (2016), and then passed through a standard feed-forward network (FFN) sub-layer. This sub-layer is likewise followed by residual connections and layer normalization, in alignment with the original Transformer architecture.

$$H^{(l)} = \textbf{LN}\left(H' + \textbf{FFN}^{(l)}(H')\right) \quad H' = \textbf{LN}\left(H^{(l-1)} + \tilde{H}^{(l)}\right) \tag{8}$$

where $H^{(l)} \in \mathbb{R}^{|V| \times d}$ denotes the output of the $l$-th encoder layer. After $L$ layers of gated fusion and Transformer blocks, the the final modality-specific node embeddings are denoted as $H_{\text{dis}}^{(L)}$ and $H_{\text{sol}}^{(L)}$. Then, the fused node embeddings are obtained by concatenating the final-layer outputs:

$$H_{\text{fuse}} = \text{Concat}\left( H_{\text{dis}}^{(L)}, H_{\text{sol}}^{(L)} \right) \tag{9}$$

### 3.3.3 FINAL STATE REPRESENTATION

To construct the final state representation, we perform mean pooling over the fused node embeddings to obtain a graph-level feature vector. This pooled representation is then concatenated with the optimization-based features to form a Unified Representation.

### 3.4 POLICY $\pi_\theta$: DECISION MODULE

After obtaining the final state representation from the encoder, we feed it into a lightweight decision module to produce the action distribution over candidate operators. Specifically, as illustrated in Fig. 1, the decision module consists of two fully connected (FC) layers to produce a vector of action probabilities. In our work, we adopt the proximal policy optimization Schulman et al. (2017) algorithm to learn the policy $\pi_\theta$.

## 4 EXPERIMENTS

In this section, we perform an in-depth analysis of the experimental results to assess its performance across various problem sizes.

### 4.1 SETUP

As recommended, we generate three benchmark datasets with different problem sizes, where the number of customers $N \in \{20, 50, 100\}$. Each instance consists of a depot and $N$ customers, all located within a two-dimensional Euclidean space $[0, 1]^2$. Customer locations are sampled uniformly at random. Demands for each customer are independently drawn from the set $\{1, 2, 3, ..., 9\}$, the vehicle capacities are set to 20, 40, and 50, when $N = 20, 50, 100$ respectively. For our GAMA, the initial solution $\delta_0$ is randomly generated. Table 5 in the appendix gives the parameter settings of the proposed GENIS, including the GNN model architecture and other algorithm parameters. In our experiments, evaluation was conducted on 500 unseen instances, and the performance was measured by the average total distance across all test cases. The training time of our GAMA varies with problem sizes, i.e., around 1 day for $N = 20$, 3 days for $N = 50$, and 7 days for $N = 100$.

### 4.2 COMPARED ALGORITHMS

To comprehensively assess the effectiveness of our proposed method, we compare it against a diverse set of baseline algorithms, including classical solvers, learning-based construction methods, and learning-based improvement methods. These baselines represent the current state of the art in both traditional and neural combinatorial optimization for CVRP. (1) Classical Heuristic and Metaheuristic Solvers, including LKH3 Helsgaun (2019), HGS Vidal (2022), and VNS Amous et al. (2017). (2) learning to construct methods, including POMO Kwon et al. (2020) and LEHD Luo et al. (2023), ReLD Huang et al. (2025). (3) Learning to improve methods, including L2I Lu et al. (2019), DACT Ma et al. (2021) and GIRE Ma et al. (2023). To evaluate the contribution of the self-and-cross attention mechanism, we compare our GAMA encoder with GENIS Guo et al. (2025).

Each neural baseline is trained using its publicly available official implementation, with hyperparameters set according to the original paper's recommendations. Each algorithm is executed 30 times independently on each dataset. Our experiments were conducted on a server equipped with 2× AMD EPYC 7713 CPUs @ 2.0GHz and 2× NVIDIA A100 GPU cards.

### 4.3 RESULTS AND DISCUSSIONS

The result is average total distance over 500 test instances, which is calculated as Eq.equation **??**, and the value is the smaller the better. Table 1 presents the performance comparison of all algo-

Table 1: Comparison results for solving CVRP instances of sizes: $|V|$ = 20, 50, and 100.

| | CVRP20 | | | CVRP50 | | | CVRP100 | | |
|---|---|---|---|---|---|---|---|---|---|
| | Best Cost | Avg. Cost | Time | Best Cost | Avg. Cost | Time | Best Cost | Avg. Cost | Time |
| LKH3 | 6.0867 | | 14s | 10.3879 | | 53s | 15.6752 | | 1.95m |
| HGS | 6.0807 | 6.0812 | 7s | 10.3515 | 10.3548 | 27s | 15.6590 | 15.6994 | 59s |
| VNS | 6.0827 | 6.0844 | 43s | 10.4140 | 10.4199 | 3.2m | 15.8843 | 15.8940 | 17m |
| POMO (gr.) | 6.1111 | 6.1768 | 0.98s | 10.5062 | 10.5702 | 1.5s | 15.7936 | 15.8451 | 2.7s |
| POMO (A=8) | 6.0904 | 6.1413 | 1.3s | 10.4472 | 10.4930 | 4.5s | 15.7337 | 15.7863 | 7s |
| LEHD (gr.) | 6.3823 | 6.3946 | 1.5s | 10.7617 | 10.7785 | 3s | 17.3004 | 17.3188 | 4s |
| LEHD (RRC=1000) | 6.0904 | 6.0915 | 35s | 10.4771 | 10.4856 | 1.6m | 15.8419 | 15.8514 | 4m |
| ReLD (gr.) | 6.1309 | 6.1401 | 0.06s | 10.4547 | 10.4676 | 0.1s | 15.7558 | 15.7558 | 0.21s |
| ReLD (A=8) | 6.1001 | 6.1041 | 0.09s | 10.3877 | 10.3958 | 0.25s | 15.6493 | 15.6593 | 0.72s |
| DACT (T=5k) | 6.0811 | 6.0817 | 55s | 10.3966 | 10.4038 | 1.8m | 15.7906 | 15.8030 | 2.54m |
| DACT (T=10k) | 6.0808 | 6.0813 | 2.1m | 10.3662 | 10.3735 | 3.5m | 15.7321 | 15.7410 | 9.5m |
| DACT (T=20k) | 6.0808 | 6.0811 | 4.4m | 10.3513 | 10.3542 | 11.2m | 15.6853 | 15.6925 | 19.3m |
| L2I (T=5k) | 6.0831 | 6.0864 | 27s | 10.4012 | 10.4310 | 1.1m | 15.8003 | 15.8914 | 4.6m |
| L2I (T=10k) | 6.0815 | 6.0835 | 57s | 10.3803 | 10.4006 | 2.19m | 15.7207 | 15.8008 | 9.2m |
| L2I (T=20k) | 6.0810 | 6.0820 | 1.9m | 10.3607 | 10.3787 | 4.37m | 15.6663 | 15.7334 | 18.7m |
| GAMA (T=5k) | 6.0823 | 6.0836 | 32.5s | 10.3966 | 10.4057 | 1.2m | 15.7339 | 15.7389 | 4.6m |
| GAMA (T=10k) | 6.0810 | 6.0818 | 1.1m | 10.3711 | 10.3742 | 2.3m | 15.6512 | 15.7054 | 9.5m |
| GAMA (T=20k) | **6.0806** | **6.0810** | 2.3m | **10.3512** | **10.3533** | 4.6m | **15.6178** | **15.6510** | 19m |

rithms on CVRP instances of sizes 20, 50, and 100. We report the best objective value, average objective value over 30 independent runs, and run one instance average cpu time for each method.

In the first group, we compare GAMA against classical optimization-based solvers, i.e., LKH3, HGS and VNS. While these methods remain strong baselines, especially on small-scale problems, their performance deteriorates as the problem size increases. In contrast, GAMA maintains superior solution quality across all instance sizes. In the second group, we include POMO, LEHD, and ReLD, which represent L2C methods. Although these methods offer fast inference, they struggle to reach high-quality solutions, particularly for larger instances. GAMA consistently outperforms them by leveraging operator-level adaptation and expressive state representations. The third group consists of recent L2I methods, including L2I and DACT, which are most closely related to our approach, their performance degrades as the problem scale increases. Compared with L2I, GAMA achieves lower objective values with fewer steps. Although GAMA incurs a longer inference time due to its iterative nature, this trade-off results in significantly better solution quality and more stable performance across diverse datasets.

### 4.4 ABLATION EVALUATION

To verify the effectiveness of various components within GAMA, we systematically remove or replace different elements and conduct an ablation study. All experiments were run 30 times independently.

Statistical significance is assessed using the Wilcoxon rank-sum test at a significance level of 0.05, which '↑', '↓' and '≈' denote that the algorithm is significantly worse than, better than, and is equal to GAMA, respectively.

Table 2: Effects of different encoding methods.

| | | CVRP20 | CVRP50 | CVRP100 |
|---|---|---|---|---|
| GENIS | best | 6.0807 | 10.3576 | 15.7306 |
| | mean | 6.0814 (↑) | 10.3604 (↑) | 15.7441 (↑) |
| | std | 0.0004 | 0.0018 | 0.0053 |
| GAMA_NG | best | 6.0809 | 10.3551 | 15.6897 |
| | mean | 6.0813 (↑) | 10.3590 (↑) | 15.7001 (↑) |
| | std | 0.0003 | 0.002 | 0.0042 |
| GAMA | best | **6.0806** | **10.3512** | **15.6178** |
| | mean | **6.0810** | **10.3533** | **15.6510** |
| | std | 0.0002 | 0.0012 | 0.0215 |

#### 4.4.1 EFFECTIVENESS OF SELF-AND-CROSS ATTENTION

To evaluate the contribution of the self-and-cross attention mechanism, we compare our GAMA encoder with GENIS Guo et al. (2025), which encodes the problem and solution graphs separately using dual GCNs without explicit cross-modal interaction.

As shown in Table 2, although GENIS performs acceptably on smaller instances (e.g., CVRP20 and CVRP50), but its mean performance deteriorates significantly on larger instances (CVRP100: 15.7441), likely due to its limited capacity to capture inter-graph dependencies. In contrast, GAMA leverages self-and-cross attention to model cross-modal dependencies, leading to consistent improvements across all instance sizes.

### 4.4.2 Effectiveness of the Gated Fusion Module

We further compare GAMA with its ablated version GAMA_NG, which removes the gated fusion and directly sums attended embeddings. While GAMA_NG outperforms GENIS, it still underperforms GAMA (e.g., CVRP100 mean: 15.7001 vs. 15.6510), showing that naive fusion limits expressiveness. The gating mechanism adaptively balances modal contributions, yielding better performance. We further illustrate this effect in Fig. 2. GAMA exhibits notably lower variance and better median performance across all time budgets.

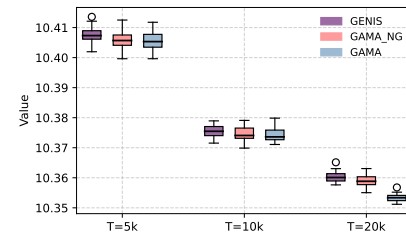

Figure 2: Solution quality distribution of GENIS, GAMA_NG, and GAMA under different inference budgets ($T = 5$k, 10k, 20k) on CVRP50.

### 4.4.3 Generalization Evaluation

We further evaluate the generalization ability of our GAMA on the classical CVRP benchmark proposed by Uchoa et al. Uchoa et al. (2017), which contains diverse instances with customer sizes ranging from 100 to 1000. To ensure a comprehensive assessment across varying scales and distributions, we systematically select several representative instances by randomly sampling. These benchmark instances exhibit substantial distributional shifts from the training set, both in terms of problem size and structural characteristics.

In Table 3, we report the best and average optimality gaps over 30 independent runs for each compared method, measured against the known optimal solutions. Detailed experimental results, including per-instance performance, are provided in the supplementary materials.

Table 3: Generalization performance on benchmark.

|       | Avg. Gap | Best Gap |
|-------|----------|----------|
| LEHD  | 9.111%   | 6.696%   |
| ReLD  | 5.018%   | 4.011%   |
| DACT  | 25.305%  | 20.527%  |
| L2I   | 13.557%  | 10.67%   |
| GAMA  | **4.956%** | **3.709%** |

Without re-training or any adaptation, GAMA achieves consistently better generalization performance than other neural baselines across all scales. This result underscores the robustness of our graph-aware multi-modal attention framework when deployed on out-of-distribution, large-scale CVRP instances.

## 5 Conclusion

In this paper, we propose GAMA, a novel Learning-to-Improve framework for the Capacitated Vehicle Routing Problem (CVRP), which formulates the operator selection process as a Markov Decision Process. By jointly encoding the problem and solution graphs through graph-aware cross- and self-attention mechanisms, and integrating their representations via a gated fusion module, GAMA effectively captures the interaction between instance structure and search dynamics. Extensive experiments on synthetic and benchmark datasets demonstrate that GAMA consistently outperforms strong neural baselines in both optimization quality and stability. Moreover, GAMA exhibits strong zero-shot generalization to out-of-distribution instances of significantly larger sizes and different spatial distributions, without retraining. In future work, we will (1)introduce data augmentation technique to further improve GAMA. (2) modeling complex operator interactions to capture dependencies and synergy among local search operators. (3) learn how to speed up the GAMA through diverse rollouts or model compression techniques.

Reproducibility Statement

All algorithmic details (c.f., Section 3 and Appendix A.3), training protocols (c.f., Section 4), and evaluation metrics (c.f., Section 4 and Section A.4) are described in the main paper and further elaborated in the Appendix. For empirical studies, we provide a detailed description of the datasets (c.f., Section 4). Hyperparameters and implementation details for all baselines are also reported in Section 4 and Appendix A.4. Upon acceptance, we will release our code and scripts for reproducing our experiments, including instructions for running data preparation. Together, these resources enable independent researchers to replicate our results and build upon our contributions.

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

## A  APPENDIX

### A.1  LEARNING-TO-OPTIMIZE METHODS FOR VRPS

L2C methods Nazari et al. (2018); Joshi et al. (2019) focus on sequentially building a feasible solution from scratch using learned policies. Despite their efficiency, these methods generally struggle to reach (near-)optimal solutions, even when enhanced with techniques Kool et al. (2018); Kim et al. (2021). Among existing construction-based approaches, POMO Kwon et al. (2020), is widely regarded as one of the most effective construction methods.

L2P methods doesn't generate solutions by itself. Instead, it predicts useful information like heuristic scores or structural properties, which are then used to guide traditional solvers or other learning components. For example, a GNN-based method was proposed Joshi et al. (2019) to predict edge-wise probability heatmaps, which are then leveraged by a beam search algorithm to solve the TSP.

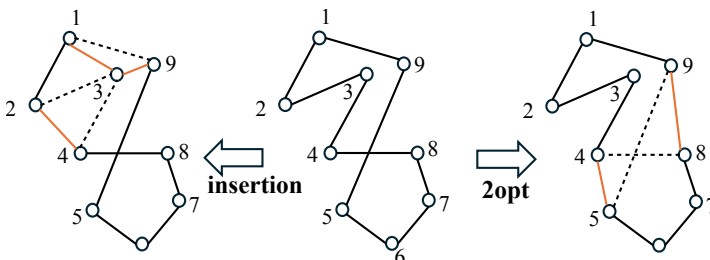

Figure 3: Illustration examples of two operators with different local optimal neighbors.

L2I methods aim to iteratively refine a given solution by modeling the search trajectory, offering a more flexible and adaptive paradigm than one-shot construction. L2I approaches can be broadly categorized into two paradigms based on how they integrate local search operations. The first paradigm adopts a fixed-operator framework, where a specific operator—such as 2-opt, relocate, or insertion—is pre-defined, and the model learns to select a node pair or an edge as the target of that operator at each step Wu et al. (2021); Ma et al. (2021). Despite their effectiveness, these approaches are inherently limited by the expressiveness and flexibility of the chosen operator. In contrast, the second paradigm introduces a more general framework to select the most suitable operator from a predefined set at each step based on the current solution state Pei et al. (2024). Our work builds on the second paradigm by viewing operator selection as a high-level action space and learning a neural policy that integrates both solution information and operator dynamics.

### A.2 PROBLEM FORMULATION

We formulate the Vehicle Routing Problem (VRP) as a combinatorial optimization problem defined over a graph $G = (V, E)$, where each node $v_i \in V$ denotes a customer or depot, and edges encode the travel cost between nodes. The objective is to minimize the total travel distance $f(\delta)$ under certain problem-specific constraints. A feasible solution $\delta$ consists of multiple sub-routes, each corresponding to a single vehicle tour. Each route starts and ends at the depot, and visits a subset of customers exactly once. The total demand served in each route must not exceed a predefined vehicle capacity $Q$. Thus, the solution must satisfy the following constraints: (1) each customer is visited exactly once, and (2) the sum of demands in each route does not exceed the vehicle capacity $Q$.

In this study, we follow the common benchmark setup proposed by Uchoa et al., for each CVRP instance, the coordinates of all nodes (customers and depot) are sampled uniformly within the unit square $[0, 1]^2$, and customer demands are drawn uniformly from $\{1, 2, ..., 9\}$. The vehicle capacity $Q$ is set to 30, 40, and 50 for CVRP20, CVRP50, and CVRP100 respectively, controlling the number of required sub-routes and the problem's combinatorial complexity.

Therefore, each VRP instance provides static problem features (e.g., node location and demand), while each candidate solution induces dynamic solution features that encode the current routing configuration and neighborhood context. Starting from an initial yet feasible solution, our learning-based AOS framework employs a neural policy to iteratively improve the solution. At each decision step, the policy selects an operator from a predefined set of local search heuristics. The details of the heuristics are presented in Table 4. The majority of the heuristics employed are canonical operators frequently used in VRP and TSP, with their effectiveness extensively validated in prior work. Once an operator is selected, it is applied exhaustively in the neighborhood of the current solution, the best improving move is then adopted to update the solution. This process is shown in Algorithm 2. Given the same input solution, different operators may yield a different locally optimal neighbor. As illustrated in Figure 3, the insert operator repositions a node, the 2-opt operator reverses a path segment. These operators explore different regions of the solution space, and their performance varies dynamically during the search, which highlights the complementarity among operators and the importance of learning to select the most suitable one at each search step.

---

**Algorithm 2:** $\delta' \leftarrow$ LocalSearch $(\delta, a)$

---

**Input** : Solution $\delta$, selected local search operator $a$, neighborhood set $\Gamma = \{\mathcal{N}_1, ..., \mathcal{N}_k, ...\}$
**Output:** Improved solution $s'$
1  $\delta' = null$, Cost $(\delta') = \infty$
2  **for** $\delta'' \in \mathcal{N}_a$ **do**
3  $\quad$ Cost $(\delta'') =$ Evaluate $(\delta'')$
4  $\quad$ **if** $Cost\,(\delta'') < Cost\,(\delta')$ **then**
5  $\quad\quad$ $\delta' \leftarrow \delta''$
6  $\quad\quad$ Cost $(\delta') \leftarrow$ Cost $(\delta'')$
7  **return** Improved solution $\delta'$

---

Table 4: Descriptions of Local Search Operators

| Type | Name(#operated routes) | Description |
|---|---|---|
| Intra-route | 2-opt(1) | Reverses a sub-route. |
| | relocate(1) | Move a segment of $m$ nodes ($m = 1, 2, 3$) in the route to a new location. |
| | swap(1) | Exchange two nodes in the same route. |
| | or-opt(1) | Replace 3 arcs with 3 new arcs in a route. |
| Inter-route | cross(2) | Exchange the segments from two routes. |
| | symmetric-swap(2) | Exchange segments of length $m$ ($m = 1, 2, 3, 4$) between two routes. |
| | asymmetric-swap(2) | Exchange segments of length $m$ and $n$ ($m = 1, 2, 3, 4, n = 1, 2, 3, 4, m \neq n$) between two routes. |
| | relocate(2) | Move a segment of length $m$ ($m = 1, 2, 3$) from a route to another. |
| | 2opt(2) | Remove two edges and reconnect their endpoints in different routes. |
| | or-opt(2) | Replace 3 arcs with 3 new arcs from another route. |
| | cyclic-exchange(3) | Exchange cyclically one customer between three routes. |

### A.3 FEATURE REPRESENTATIONS

In our neural neighborhood search framework for solving CVRP, the state at decision step $t$, denoted as $s_t$, is designed to comprehensively represent the current search context. Specifically, it integrates structural features from the problem and current solution, as well as relevant statistics from the optimization history. The complete state feature is formulated as:

$$s_t = \{\mathcal{G}_{\text{dis}}, \mathcal{G}_{\text{sol}}, \mathcal{X}_t, a, e, \Delta, \eta\} \tag{10}$$

Each component is described as follows:

- Distance Graph $\mathcal{G}_{\text{dis}}$: A fully connected graph encoding the spatial structure of the problem instance. Each node corresponds to a customer or depot, and edge weights represent Euclidean distances between node pairs, given by $\mathcal{G}_{\text{dis}} = \|\text{loc}(i) - \text{loc}(j)\|_2 = \sqrt{(x_i - x_j)^2 + (y_i - y_j)^2}$. This graph is static across the entire search process and captures the underlying geometry of the problem.

- Solution Graph $\mathcal{G}_{\text{sol}}$: A subgraph dynamically induced by the current routing solution. For any two nodes $i$ and $j$, $\mathcal{G}_{\text{sol}}[i, j] = 1$ if they are directly connected in the current routing solution, and 0 otherwise. This graph evolves over time as the solution is updated and provides insight into the local neighborhood and tour connectivity.

- Node Features $\mathcal{X}_t$ at Time $t$: A matrix of node-level features that encode both static and dynamic attributes of each node. Each node $i$ comprises 11 features, which include the

Table 5: Parameter Setting of GAMA.

|  | Description | Value |
|---|---|---|
| GCN | number of GCN layers | 2 |
| | GCN output hidden dimension | 16 |
| | GCN activation functions | ReLU |
| | MLP output hidden dimension | 32 |
| Attention | Head number of attention | 4 |
| | Input hidden dimension of each head | 16 |
| | Output hidden dimension of self-attention | 32 |
| other | learning rate | 3e-4 |
| | Optimizer to training neural networks | ADAM |
| | Maximum episodes $NoE$ | 500 |
| | Maximum timesteps $T$ | 20000 |
| | max no improvement threshold $L$ | 6 |

two-dimensional coordinate, the customer's demand $q_i$, the remaining capacity after the vehicle arrives at this node, the coordinates of its two adjacent neighbors in the current routing solution (the predecessor $i^-$ and successor $i^+$), and the Euclidean distances between the node and its adjacent nodes, specifically $\|i - i^-\|_2$, $\|i - i^+\|_2$, and $\|i^- - i^+\|_2$;

- Optimization-based Features: including last applied operator $a$; its improvement effectiveness $e$ to measure the solution improvement. If the last action successfully improves the current solution, $e = 1$; otherwise, $e = 0$; the gap between the current solution and the current best solution $\Delta$; and the change in objective value caused by the last action $\eta$.

### A.4   More discussion on the experiments

The parameter settings of the proposed GAMA is given in Table 5, including the GNN model architecture and other algorithm parameters.

#### A.4.1   Convergence Analysis

To further understand the inference dynamics, we plot the inference-time convergence trajectories of GAMA, L2I, and DACT on CVRP20/50/100 in Figure 4. Each curve corresponds to a single representative run and shows how solution quality evolves with increasing rollout steps.

Across all instance sizes, we observe that:

- In the early phase (up to Rollout Steps = 2500), all methods exhibit comparable performance, making steady improvements as search proceeds.

- Beyond 2500 steps, however, DACT and L2I quickly reach a plateau, and their improvement slows down very slowly - this is especially evident in CVRP100, where both methods stagnate prematurely.

- In contrast, GAMA continues to improve steadily across the entire inference horizon, regardless of problem scale. This indicates that GAMA's attention-based encoder and adaptive fusion enable it to effectively explore deeper and more promising neighborhoods during late-stage rollouts.

These trends suggest that GAMA maintains better long-term optimization ability, avoiding early convergence.

#### A.4.2   Detailed Analysis on Encoder Effectiveness

To provide a more comprehensive evaluation of our encoder design, we report in Table 6 the performance of three encoder variants—GENIS, GAMA_NG, and the proposed GAMA—under different training budgets ($T = 5k, 10k, 20k$ steps). The results reveal consistent and significant trends across all instance sizes (CVRP20, CVRP50, and CVRP100).

At all training steps, GAMA outperforms or performs equal to the other two baselines in both best and average solution quality, with especially clear advantages on the largest and most challeng-

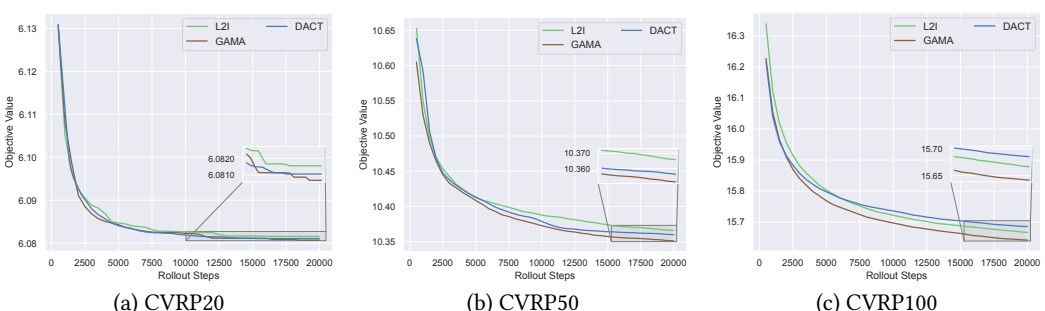

(a) CVRP20       (b) CVRP50       (c) CVRP100

Figure 4: Convergence curves of GAMA and different L2I methods.

Table 6: Experimental details of encoder effectiveness.

|  |  | CVRP20 | CVRP50 | CVRP100 |
|---|---|---|---|---|
| GENIS | best | 6.0825 | 10.4020 | 15.8344 |
| (T=5k) | mean | 6.0839 (↑) | 10.4075 (↑) | 15.8503 (↑) |
|  | std | 0.0008 | 0.0027 | 0.0057 |
| GAMA_NG | best | 6.0824 | **10.3996** | 15.8354 |
| (T=5k) | mean | 6.0835 (≈) | 10.4058 (≈) | 15.8452 (↑) |
|  | std | 0.0005 | 0.0028 | 0.0042 |
| GAMA | best | **6.0823** | **10.3996** | **15.7105** |
| (T=5k) | mean | **6.0836** | **10.4057** | **15.7768** |
|  | std | 0.0006 | 0.0031 | 0.0340 |
| GENIS | best | 6.08011 | 10.3715 | 15.7744 |
| (T=10k) | mean | **6.0818** (≈) | 10.3755 (↑) | 15.7834 (↑) |
|  | std | 0.0003 | 0.0020 | 0.0045 |
| GAMA_NG | best | 6.0811 | **10.3699** | 15.7693 |
| (T=10k) | mean | 6.0816 (↑) | 10.3744 (≈) | 15.7780 (↑) |
|  | std | 0.0003 | 0.0024 | 0.0039 |
| GAMA | best | **6.0810** | 10.3711 | **15.6512** |
| (T=10k) | mean | **6.0818** | **10.3742** | **15.7054** |
|  | std | 0.0005 | 0.0021 | 0.0280 |
| GENIS | best | 6.0807 | 10.3576 | 15.7306 |
| (T=20k) | mean | 6.0814 (↑) | 10.3604 (↑) | 15.7441 (↑) |
|  | std | 0.0004 | 0.0018 | 0.0053 |
| GAMA_NG | best | 6.0809 | 10.3551 | 15.6897 |
| (T=20k) | mean | 6.0813 (↑) | 10.3590 (↑) | 15.7001 (↑) |
|  | std | 0.0003 | 0.002 | 0.0042 |
| GAMA | best | **6.0806** | **10.3511** | **15.6178** |
| (T=20k) | mean | **6.0810** | **10.3533** | **15.6510** |
|  | std | 0.0002 | 0.0012 | 0.0215 |

ing instances (e.g., CVRP100). For example, under $T = 20k$, GAMA achieves an average cost of 15.6510, which improves over 15.7001 from GAMA_NG and 15.7441 from GENIS. This highlights the scalability of our encoding strategy.

**Effectiveness of self-and-cross attention:** GENIS utilizes dual GCNs to encode the problem and solution graphs independently, followed by simple concatenation and a shallow self-attention module. This design leads to limited interaction between the two modalities. Under small training budgets (e.g., $T = 5k$), GENIS performs comparably to GAMA_NG and even GAMA on small problems (CVRP20), but significantly lags on CVRP50 and CVRP100. As training increases, GENIS

Table 7: Performance Comparison on CVRP Library Benchmarks

|  |  | BKS | LEHD | ReLD | L2I | DACT | GAMA |
|---|---|---|---|---|---|---|---|
| X-n101-k25 | best | 27591 | 28145.61 | 28033 | 28076.85 | 29359.58 | 27950.24 |
|  | mean |  | 28386.42 | 28237.73 | 28417.64 | 32212.39 | 28204.41 |
|  | std |  | 130.69 | 125.06 | 179.16 | 1780.99 | 106.72 |
|  | A. G. |  | 2.883% | 2.344% | 2.996% | 16.749% | 2.223% |
| X-n261-k13 | best | 26558 | 27514.34 | 27320 | 28608.15 | 33860.63 | 27696.61 |
|  | mean |  | 28013.16 | 27381.3 | 29250.14 | 35969.73 | 28141.57 |
|  | std |  | 258.51 | 44.94 | 275.68 | 951.49 | 154.02 |
|  | A. G. |  | 5.479% | 3.1% | 10.137% | 35.438% | 5.963% |
| X-n331-k15 | best | 31102 | 32336.99 | 32027 | 33359.79 | 35807.18 | 32378.05 |
|  | mean |  | 33020.61 | 32115.3 | 33839 | 36305.12 | 32598.64 |
|  | std |  | 299.57 | 55.75 | 307.24 | 145.14 | 124.18 |
|  | A. G. |  | 6.168% | 3.257% | 8.8% | 16.729% | 4.812% |
| X-n420-k130 | best | 107798 | 119403.97 | 110462 | 110092.01 | 127712.69 | 109654.01 |
|  | mean |  | 120437.96 | 110730.9 | 111104.40 | 129816.05 | 110299.95 |
|  | std |  | 628.52 | 161.03 | 325.61 | 1235.13 | 234.29 |
|  | A. G. |  | 11.725% | 2.72% | 3.067% | 20.425% | 2.321% |
| X-n513-k21 | best | 24201 | 27052.21 | 25597 | 26462.71 | 31143.84 | 25099.95 |
|  | mean |  | 27532.85 | 25987.7 | 27273.49 | 31638.12 | 25918.88 |
|  | std |  | 250.29 | 62.55 | 501.66 | 96.40 | 400.39 |
|  | A. G. |  | 13.767% | 7.382% | 12.695% | 30.730% | 7.098% |
| X-n613-k62 | best | 59535 | 65013.57 | 62287 | 64333.62 | 78296.5 | 62003.09 |
|  | mean |  | 65701.16 | 63042.7 | 65365.06 | 79951.56 | 62956.56 |
|  | std |  | 286.52 | 282.08 | 591.02 | 1082.35 | 454.34 |
|  | A. G. |  | 10.357% | 5.892% | 9.792% | 34.293% | 5.747% |
| X-n701-k44 | best | 81923 | 88488.54 | 85197 | 92034.43 | 92940.80 | 85358.34 |
|  | mean |  | 89316.59 | 86156.53 | 94453.01 | 93177.37 | 85904.55 |
|  | std |  | 421.9 | 801.91 | 1056.73 | 95.917 | 554.27 |
|  | A. G. |  | 9.0250% | 5.167% | 15.294% | 13.737% | 4.86% |
| X-n801-k40 | best | 73311 | 79154.95 | 76682 | 88188.46 | 111656.89 | 76043.89 |
|  | mean |  | 80290.66 | 77037.4 | 93258.41 | 120036.77 | 76701.02 |
|  | std |  | 442.79 | 135.11 | 5127.52 | 3621.25 | 611.32 |
|  | A. G. |  | 9.521% | 5.08% | 27.209% | 63.736% | 4.624% |
| X-n916-k207 | best | 329179 | 354583.06 | 340357 | 363036.36 | 347893.41 | 336099.09 |
|  | mean |  | 357081.81 | 341262.46 | 368784.99 | 356082.14 | 336538.92 |
|  | std |  | 946.27 | 388.73 | 2383.99 | 4638.04 | 278.33 |
|  | A. G. |  | 8.4764% | 3.67% | 12.031% | 8.172% | 2.236% |
| X-n1001-k43 | best | 72355 | 80981.09 | 78426.43 | 92196.18 | 76787.63 | 77996.14 |
|  | mean |  | 82048.95 | 80726.83 | 96627.31 | 81789.21 | 79359.78 |
|  | std |  | 551.22 | 486.75 | 1961.46 | 2558.03 | 705.39 |
|  | A. G. |  | 13.397% | 11.57% | 33.546% | 13.039% | 9.681% |
| Total Avg. Gap |  |  | 9.111% | 5.018% | 13.557% | 25.305% | **4.956%** |

does improve, but at a slower rate than GAMA. Even at $T = 20k$, its average performance remains worse than both GAMA_NG and GAMA, suggesting an architectural limitation.

**Effectiveness of the Gated Fusion Module:** GAMA_NG incorporates the same self-and-cross attention encoder as GAMA but removes the gated fusion, using simple addition for feature merging. This preserves more modal interaction than GENIS but lacks adaptive control over information flow. In early training stages, GAMA_NG achieves slightly better performance than GENIS, especially on medium/large instances (e.g., CVRP100 with $T = 5k$), indicating that cross-attention already contributes to better representation. However, the lack of adaptive gating makes it harder to balance the importance of problem vs. solution graph features, especially when solution structures become complex.

**Effect of Training Steps:** We also observe that performance improves steadily with more training. For all methods, increasing $T$ from $5k$ to $20k$ reduces the cost across all problem sizes, indicating that sufficient training is crucial for model effectiveness. However, GAMA consistently maintains its lead at every training step, which supports the claim that its architectural design—not just training duration—plays a key role in achieving high solution quality.

In conclusion, the results clearly validate both components of our encoder design: the self-and-cross attention mechanism, which enables explicit cross-modal interaction, and the gated fusion module, which adaptively integrates problem and solution embeddings. These components jointly contribute to GAMA's superior performance and generalization ability.

### A.4.3 Generalization on benchmark datasets

We further evaluate the generalization ability of our proposed GAMA framework on the well-established CVRPLib benchmark suite introduced by Uchoa et al. Uchoa et al. (2017), which consists of diverse real-world-inspired CVRP instances with customer sizes ranging from 100 to 1000. To ensure a representative and challenging evaluation, we systematically select 10 instances with varying size, vehicle count, and spatial distribution characteristics, thereby inducing substantial distribution shifts from our training set.

All baselines, including LEHD, ReLD, L2I, and DACT, are evaluated using their official implementations. As shown in Table 7, our GAMA achieves the lowest total average optimality gap of 4.96%, outperforming LEHD (9.11%), ReLD (5.018%), L2I (13.56%), and DACT (25.31%) by substantial margins. GAMA consistently delivers competitive or superior best and mean solutions across almost all instances, especially on larger and more complex cases such as X-n801-k40 and X-n916-k207. This evidences its strong out-of-distribution generalization ability and robustness to scale variation. Notably, DACT exhibits significantly inferior performance on large-scale benchmarks. This can be largely attributed to: it employs a fixed local search operator (2-opt) throughout optimization, limiting its adaptability to diverse problem structures. Although GAMA incurs a higher average inference time compared to L2I and DACT, this additional cost is offset by its substantially improved solution quality. For high-stakes logistics applications, such a trade-off is often desirable.

These results collectively demonstrate that GAMA generalizes robustly to a wide variety of real-world CVRP scenarios, thanks to its expressive graph-based state representation and adaptive operator selection policy.

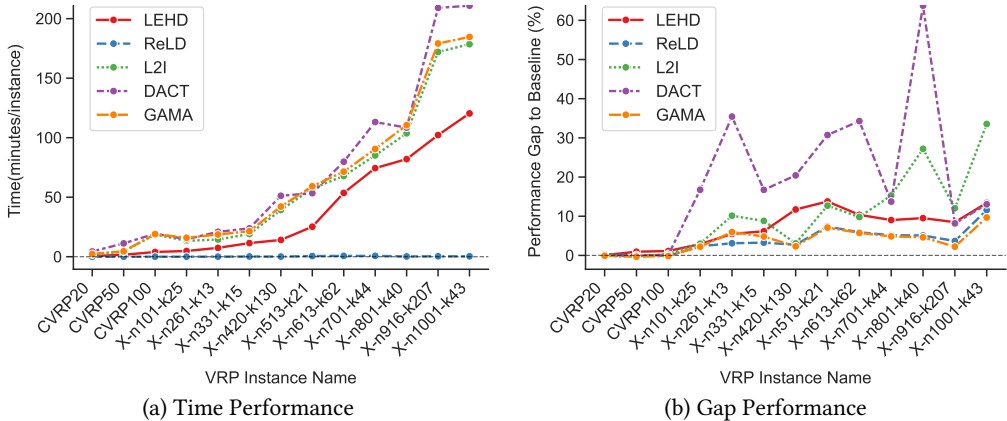

Figure 5: Performance comparison between different methods on VRP instances. Left: Computation time. Right: Performance gap to baseline.

Table 8: Comparison results for solving CVRP instances of sizes: $|V|$ = 20, 50, and 100.

| | CVRP1000 | | | CVRP2000 | | |
|---|---|---|---|---|---|---|
| | Best | Avg | Time | Best | Avg | Time |
| POMO | 93.8710 | 143.5383 | 30s | 396.3919 | 486.5468 | 80s |
| POMO(A=8) | 59.1911 | 87.6515 | 3.2m | 169.8233 | 285.5397 | 43m |
| LEHD | 48.5214 | 46.2542 | 32s | 158.65 | 149.54 | 8m |
| LEHD(RRC=1000) | 37.6219 | 38.5661 | 2.3h | 117.52 | 122.4589 | 5.9h |
| ReLD | 38.7953 | 39.0602 | 8s | 63.3600 | 63.6378 | 33s |
| ReLD(A=8) | 38.4513 | 38.6234 | 44s | 61.9754 | 62.3018 | 2.92m |
| DACT(T=5k) | 45.7543 | 46.3442 | 4h | 70.6339 | 77.4895 | 10h |
| DACT(T=10k) | 44.7730 | 45.7689 | 8.1h | 70.6339 | 72.5638 | 20.5h |
| DACT(T=20k) | 44.1439 | 45.0561 | 16h | 68.1593 | 69.7607 | 41h |
| L2I(T=5k) | 47.0318 | 68.431 | 18m | 74.3857 | 153.071 | 1.5h |
| L2I(T=10k) | 45.1571 | 62.1688 | 36m | 70.3181 | 132.6643 | 3h |
| L2I(T=20k) | 45.1571 | 57.872 | 1.2h | 70.3181 | 116.3726 | 6h |
| GAMA(T=5k) | 36.9435 | 37.2608 | 22.5m | 58.5433 | 60.0774 | 1.8h |
| GAMA(T=10k) | 36.7561 | 37.0043 | 44.8m | 57.9618 | 58.7187 | 3.6h |
| GAMA(T=20k) | **36.7561** | **36.7768** | 1.5h | **57.9618** | **58.0593** | 7.2h |

### A.4.4 PERFORMANCE ANALYSIS

We also conducted a comprehensive performance evaluation of all compared methods, summarized in Figure 5. The results strongly validate the superior capability of the GAMA approach in solving VRPs.

- **Superior Solution Quality**: As depicted in Figure 5 (b), GAMA almostly achieved the best solution quality (smallest gap to the baseline) across all tested instances. Notably, on the small CVRP20, CVRP50, and CVRP100 instances, where LKH3 serves as the baseline, GAMA achieved an average gap of $-0.194\%$, making it the only method to outperform the LKH3 baseline on average and on every size dataset. This confirms GAMA's high search efficiency, even against strong exact heuristics on small-to-medium problems. Furthermore, on the larger CVRPLIB instances (benchmarked against BKS), GAMA's average gap is significantly lower than all comparative methods, demonstrating excellent robustness and generalization on large-scale VRP instances.

- **Balanced Performance and Efficiency**: As shown in Figure 5 (a), despite GAMA's complex search strategy aimed at optimizing solution quality, its computational overhead remains acceptable. GAMA's average running time is on par with other L2I methods (such as DACT, L2I). It is reasonable and acceptable to spend some time to improve quality.

### A.4.5 EVALUATION ON LARGE-SCALE VRP

We further evaluated the performance of our proposed GAMA method on large-scale VRP instances (CVRP1000, Capacity=250; CVRP2000, Capacity=300), following the suggestions by Luo et al. (2025). The evaluation results are presented in Table 8. The analysis demonstrates that GAMA maintains the best solution quality, even when faced with large-scale problems. Across both CVRP1000 and CVRP2000 datasets, GAMA(T=20k) consistently achieved the Best and Avg results among all comparative methods. For instance, on CVRP1000, GAMA(T=20k)'s average tour length was 36.7768, significantly outperforming the next best ReLD; similarly, on CVRP2000, the average solution length substantially surpassed other L2Opt methods' performance. This strongly validates GAMA's generalization ability and robustness for tackling large-scale VRP instances.

## LLM USAGE STATEMENT

We used ChatGPT (GPT-5) only as an assistive tool for grammar checking and language polishing. The model was not involved in research ideation, algorithm design, experiment execution, or result analysis. All scientific content and conclusions are entirely the work of the authors.

