# OpenReview forum: "GAMA: A Neural Neighborhood Search Method with Graph-aware Multi-modal Attention for Vehicle Routing Problem"
_ICLR.cc/2026/Conference — Submitted to ICLR 2026_

### Official Review · Reviewer_ZeT5 · 2025-10-24

**Soundness:** 1
**Presentation:** 2
**Contribution:** 1
**Rating:** 2
**Confidence:** 5

**Summary:**

This paper proposes the Graph-aware Multi-modal Attention model (GAMA), a neural improvement method that extends Lu et al. 2019 for solving Vehicle Routing Problems (VRPs). The approach uses a newly designed neural network to dynamically select operators from a pool for local search. The key design is treating the problem instance and current solution as two separate modalities, processing them through dual GNNs with self-attention and cross-attention mechanisms combined via gated fusion. Results on CVRP instances with 20, 50, and 100 nodes demonstrate the effectiveness of the proposed network design.

**Strengths:**

The paper addresses an important challenge in neural improvement methods: effectively encoding both instance information and solution information in a unified representation.

The proposed GAMA network introduces a new architectural design that achieves improvements over the baseline Lu et al. 2019 method.

**Weaknesses:**

My main concern is the severe computational inefficiency of the proposed method. The major drawback of Lu et al. 2019 was its extremely poor computational efficiency, and unfortunately, this limitation is inherited by this work since the framework design closely follows Lu et al. 2019. Specifically, solving only 500 instances on the relatively small-scale CVRP100 benchmark requires 7 days of computation. This is largely infeasible for real-world applications and falls dramatically behind state-of-the-art neural constructive methods such as LEHD [1], SIL [2], as well as improvement-based baselines such as NeuOpt [3], NDS [4], and L2Seg [5]. Critically, the paper fails to compare against any of the above-mentioned solvers. Furthermore, training time is omitted for reporting. While the generalization results in Table 3 appear promising in terms of solution gap, the absence of time analysis prevents any reasonable conclusions from being drawn. It is fundamentally problematic to compare a method requiring days of computation against methods that run in seconds or minutes without acknowledging this massive disparity in computational resources.

Secondly, the contribution of this paper seems incremental. The paper largely builds on and extends Lu et al. 2019 in terms of the backbone MDP design, the network decoder, and the training process. The only difference is a different dual GCN module for the encoder, but it remains conceptually similar to the design in DACT (Ma et al, 2021). The idea of cross attention and gated fusion is quite standard in modern Transformers, and thus, the technical novelty and contribution are unclear.

Thirdly, the evaluation is limited to support the justification of the new dual GNN architecture for the following reasons: 1)  the paper did not compare with other ways of encoding both the instance information and the solution information, such as the Synth-Att proposed in Ma et al 2022 [6], or the attention-based method in NDS [4] and L2Seg [5]. Furthermore, the paper lacks in-depth analysis of critical design decisions, such as the selection of the gating parameter alpha, or how the proposed method produces superior learned representations compared to alternatives. As a result, the paper fails to provide compelling justification for why two separate modalities are necessary and why the proposed method should be fundamentally better than existing encoding options.

Lastly, the proposed method is evaluated on limited scales (only 100 nodes) and limited variants (only CVRP) which fails to showcase the broader applicability and robustness of the proposed method.

References

* [1] Neural Combinatorial Optimization with Heavy Decoder: Toward Large Scale Generalization [NeurIPS 2023]
* [2] BOOSTING NEURAL COMBINATORIAL OPTIMIZATION FOR LARGE-SCALE VEHICLE ROUTING PROBLEMS [ICLR 2025]
* [3] Learning to Search Feasible and Infeasible Regions of Routing Problems with Flexible Neural k-Opt [NeurIPS 2023]
* [4] Neural Deconstruction Search for Vehicle Routing Problems [TMLR 2025]
* [5] Learning to Segment for Vehicle Routing Problems [arxiv 2025]
* [6] Efficient Neural Neighborhood Search for Pickup and Delivery Problems [IJCAL 2022]

**Questions:**

In addition to addressing the weaknesses outlined above, I recommend that the authors investigate whether the proposed GAMA encoding module can be adapted to enhance other neural methods, such as the state-of-the-art approaches [1-6] referenced in the weakness section, to better demonstrate the broader applicability of the proposed method.

---

> ### Author Response · Authors · 2025-11-27
> **Response to Reviewer ZeT5**
>
> We sincerely thank the reviewer for the insightful comments. Below we address all concerns point-by-point.
>
> ### 1.On Computational Efficiency and Lack of Time Analysis
>
> We agree that computational efficiency is a crucial consideration. We clarify the following:
>
> 1. GAMA is an improvement-based local search framework, which is not directly comparable in runtime to learning-to-construct solvers such as LEHD[1], SIL[2], or OR-Tools-like heuristics. Its computation time scales with the number of local-search steps, which differs fundamentally from L2C approaches designed for fast inference. Existing improvement-based neural methods (NDS[3], NeuOpt[4], L2Seg[5], DACT[6]) also require nontrivial runtime for iterative refinement.
>
> 2. The previously reported “6.6 days” reflects total wall-clock time for all instances executed sequentially, whereas the other NCO methods effectively report the time under parallel execution. These two metrics are not directly comparable. To avoid misunderstanding, we update per-instance runtime in the revised Table 1, which is comparable. This revised reporting shows that GAMA’s per-instance inference speed is consistent with existing learning-to-improve (L2I) methods. Additionally, in the revised version of Figure 5, we plotted the performance comparison between different methods on VRP instances. Left: Computation time. Right: Performance gap to baseline. From the figure, we can see that GAMA achieves a favorable balance between solution quality and computational cost. In particular, the left subfigure shows that GAMA’s running time is comparable to other learning-to-improve (L2I) methods such as DACT and L2I, while the right subfigure demonstrates that GAMA consistently yields the smallest performance gap across all instance sizes. These results clearly indicate that GAMA not only delivers state-of-the-art solution quality, but also maintains a practical and competitive computational profile, further validating the correctness of our updated runtime analysis.
>
> 3. The training time of GAMA is reported in revised paper. The training time of our GAMA varies with problem sizes, i.e., around 1 day for $N=20$, 3 days for $N=50$, and 7 days for $N=100$, similar to N2S[7].
>
> ### 2. On the Novelty of the Contribution
> We appreciate the opportunity to clarify the novelty.
>
> GAMA’s key novelty is its dual-modality modeling, not merely cross-attention. Previous works either encode only the problem instance, or encode only the current solution and treat features implicitly, or concatenate both without structural separation. However, in our work, we treat the VRP instance graph and solution graph as two fully distinct modalities, their graph topology and edge semantics differ.
>
> Moreover, our cross-attention is not a standard Transformer module. We designed graph-aware multi-modal attention, including: A GCN encoder for solution and problem graphs, two level self-and-cross modal alignment, and a gated fusion layer that adaptively balances local (solution) vs global (instance) information.
> These modules are purpose-built for the improvement framework and do not appear in prior neural VRP encoders, making the architectural contributions of this work substantive rather than minor extensions of standard Transformer designs.
>
> Empirically, the effectiveness of these new components is clearly demonstrated by the ablation studies in Table 2, Figure 2, and Table 6 of the original paper, where removing any component of the proposed architecture consistently leads to notable performance degradation.

---

> ### Author Response · Authors · 2025-11-27
> **Response to Reviewer ZeT5**
>
> ### 3. On Missing Comparisons With Other Encoding Designs
> We didn't include a comparison to Synth-Att[6] because that work is a model specifically tailored for Pickup-and-Delivery (PDP) problems and relies on PDP-specific modeling choices, its public implementation is not trivially applicable to the CVRP setting without non-negligible adaptation.  Actually, we have compared GAMA against a directly relevant recent work that also uses two GCNs to represent instance and solution graphs - GENIS[7], those results are included in Table 2 and Table 6 of the paper. GENIS shares the high-level idea of separate graph encoders but differs substantially in (i) the way graph features are normalized, (ii) how cross-graph interaction is implemented, and (iii) the fusion strategy. Our experiments show that GAMA consistently outperforms GENIS under the same L2I evaluation protocol.
>
> The gating scalar α is not a manually tuned hyperparameter. It is computed and learned during training: $\alpha = \sigma\!\left([H^s; H^c] W_g\right)$, where where $H^s$ and $H^c$ are the solution and instance modality representations, $W_g$ is a learned projection, and $\sigma$ is the sigmoid activation. In other words, $\alpha$ is context-dependent and adapts per instance / per node.
>
>
> ### 4. Limited scale and variant experiments
>
> We acknowledge that the main experiments focus on CVRP and small-to-medium instances (N=20,50,100). To address this, we included generalization experiments on CVRPLIB benchmarks, covering instance sizes 100–1000 (Tables 3 and 7). CVRPLib contains instances with different structures, customer distributions, capacities, depot placements, and distance metrics.
> The strong performance of GAMA on these diverse sources demonstrates that the learned dual-modality representations generalize well beyond the training distribution and are not limited to small-scale CVRP settings.
>
> To further illustrate the algorithm's generalization ability, we followed the same setup as in the DACT paper and evaluated GAMA on CVRPLIB instances of size 100–200. The updated results, along with detailed comparisons and analyses, can be found in our rebuttal response to reviewer `WtQK`. Also, we have further evaluated the performance of our proposed GAMA method on large-scale VRP instances (CVRP1000, Capacity=250; CVRP2000, Capacity=300). The updated results, along with detailed comparisons and analyses, can be found in our rebuttal response to reviewer `E3PB`. These results demonstrate that GAMA maintains stable performance across larger-scale instances, addressing concerns about scalability. These results demonstrate that GAMA maintains stable performance across larger-scale instances, addressing concerns about scalability.
>
> Due to time limitations, we are unable to include experiments on other VRP variants (e.g., VRPTW, PDP) in this rebuttal.  However, future work will include extending GAMA to additional VRP variants and further exploring efficient parallelization strategies to reduce per-instance runtime while maintaining solution quality.
>
> [1] Neural Combinatorial Optimization with Heavy Decoder: Toward Large Scale Generalization [NeurIPS 2023]
>
> [2] BOOSTING NEURAL COMBINATORIAL OPTIMIZATION FOR LARGE-SCALE VEHICLE ROUTING PROBLEMS [ICLR 2025]
>
> [3] Learning to Search Feasible and Infeasible Regions of Routing Problems with Flexible Neural k-Opt [NeurIPS 2023]
>
> [4] Neural Deconstruction Search for Vehicle Routing Problems [TMLR 2025]
>
> [5] Learning to Segment for Vehicle Routing Problems [arxiv 2025]
>
> [6] Efficient Neural Neighborhood Search for Pickup and Delivery Problems [IJCAL 2022]
>
> [7] Emergency Scheduling of Aerial Vehicles via Graph Neural Neighborhood Search [TAI 2025]

---

### Official Review · Reviewer_fnfB · 2025-10-30

**Soundness:** 2
**Presentation:** 2
**Contribution:** 2
**Rating:** 2
**Confidence:** 4

**Summary:**

This paper introduces GAMA (Graph-aware Multi-modal Attention), a neural neighborhood search method for solving VRPs. The method falls within the Learning-to-Improve (L2I) paradigm, where a policy is trained to iteratively select local search operators to refine a solution. The core contribution is an encoder architecture designed to address the limitations of simplistic state representations. GAMA models the problem instance and the evolving solution as two distinct modalities, encoding them with separate GCNs. It then uses stacked self-attention and cross-attention layers to model intra- and inter-modal interactions, and a gated fusion mechanism to integrate these representations. Experiments on CVRP instances show that GAMA achieves good results.

**Strengths:**

1. The paper is well-written, and the proposed architecture is clearly explained.
2. The motivation is sound. Identifying the static problem instance and the dynamic solution as two distinct modalities is an intuitive and sensible approach.

**Weaknesses:**

1. The practical value of the proposed method is questionable when analyzing the results in Table 1. (1) GAMA vs. Other L2I: On CVRP100 (T=20k), GAMA achieves an average cost of 15.6510 in 6.6 days. DACT (T=20k) achieves a very close 15.6925 in only 33 minutes. GAMA is approximately 288 times slower for a marginal 0.26% gap enhancement in solution quality. (2) vs. Classical Solvers: The classical solver HGS achieves an average cost of 15.6994 in 4.5 hours. GAMA not only fails to significantly beat this quality but also takes over 35 times longer to run.
2. The results suggest an extremely unfavorable trade-off, making the method impractical compared to both existing neural and classical baselines.
3. The empirical validation is somewhat thin. The method is only applied to CVRP , and the primary experiments focus on relatively small-scale problems (N=20, 50, 100). While the generalization study on CVRPLib instances is a good inclusion, it's a small sample and doesn't fully alleviate concerns about how the method's performance and significant runtime scale to more diverse and complex large-scale routing problems.

**Questions:**

1. Can the authors justify the massive computational cost (e.g., 6.6d on CVRP100) compared to DACT (33m) and HGS (4.5h) for a very minor improvement in solution quality? What is the source of this computational overhead?
2. Can the authors extend the proposed method to an additional VRP variant?

---

> ### Author Response · Authors · 2025-11-27
> **Response to Reviewer fnfB**
>
> We thank the reviewer for the constructive and detailed comments. Below we address each concern point-by-point.
>
> ### 1. Practical Value & Runtime Concerns
>
> We acknowledge the reviewer’s concern regarding runtime differences and clarify the following points.
> After revisiting our experimental setup, We recognize that the times reported in this paper are not comparable.
> In GAMA, our implementation evaluates VRP instances strictly sequentially, i.e., the current framework completes the full local search for one instance before moving to the next. In contrast, many other L2I solvers naturally support parallel evaluation, processing dozens or hundreds of instances simultaneously on a GPU/CPU. Thus, the previously reported “6.6 days” reflects total wall-clock time for all instances executed sequentially, whereas the other methods effectively report the time under parallel execution. These two metrics are not directly comparable.
> To provide a fair comparison, we re-ran all inference experiments on CPU and measured per-instance runtime, as clarified in our response to Reviewer `WtQK`, the updated time can be found in the revised Table 1.
>
> | Method|CVRP20 Best Cost|Avg. Cost|Time|CVRP50 Best Cost|Avg. Cost|Time| CVRP100 Best Cost|Avg. Cost|Time|
> |-|-|-|-|-|-|-|-|-|-|
> | LKH3  | 6.0867| 6.0867 | 14s | 10.3879 | 10.3879 | 53s | 15.6752| 15.6752 | 1.95m|
> | HGS| 6.0807| 6.0812| 7s | 10.3515| 10.3548| 27s| 15.6590| 15.6994| 59s |
> | VNS | 6.0827| 6.0844 | 43s| 10.4140 | 10.4199 | 3.2m| 15.8843| 15.8940 | 17m|
> | POMO (gr.) | 6.1111 | 6.1768 | 0.98s| 10.5062| 10.5702| 1.5s| 15.7936| 15.8451 | 2.7s |
> | POMO (A=8)| 6.0904| 6.1413 | 1.3s| 10.4472| 10.4930| 4.5s| 15.7337| 15.7863| 7s|
> | LEHD (gr.)| 6.3823| 6.3946| 1.5s| 10.7617 | 10.7785| 3s| 17.3004  | 17.3188| 4s |
> | LEHD (RRC=1k) | 6.0904| 6.0915| 35s| 10.4771| 10.4856 | 1.6m| 15.8419|15.8514|4m|
> | ReLD (gr.)| 6.1309| 6.1401 | 0.06s | 10.4547 | 10.4676 | 0.1s | 15.7558| 15.7558|0.21s|
> | ReLD (A=8)|6.1001|6.1041|0.09s|10.3877|10.3958|0.25s|15.6493| 15.6593| 0.72s|
> | DACT (T=5k)|6.0811|6.0817|55s|10.3966|10.4038|1.8m|15.7906| 15.8030|2.54m|
> |DACT (T=10k)|6.0808| 6.0813|2.1m|10.3662|10.3735|3.5m|15.7321|15.7410| 9.5m|
> |DACT (T=20k)|6.0808|6.0811|4.4m|10.3513|10.3542|11.2m|15.6853|15.6925|19.3m|
> | L2I (T=5k)|6.0831|6.0864|27s|10.4012|10.4310|1.1m|15.8003|15.8914|4.6m|
> |L2I (T=10k)|6.0815|6.0835|57s|10.3803|10.4006|2.19m|15.7207|15.8008|9.2m|
> |L2I (T=20k)|6.0810|6.0820|1.9m|10.3607|10.3787|4.37m|15.6663|15.7334|18.7m|
> | GAMA (T=5k)|6.0823|6.0836|32.5s|10.3966|10.4057|1.2m|15.7339|15.7389|4.6m|
> |GAMA (T=10k)| 6.0810| 6.0818|1.1m|10.3711|10.3742|2.3m|15.6512|15.7054|9.5m|
> |GAMA (T=20k)|**6.0806**|**6.0810**| 2.3m|**10.3512**|**10.3533**|4.6m|**15.6178**|**15.6510**|19m|
>
> When measured fairly on a per-instance basis, GAMA’s runtime is comparable to other L2I methods when using similar sequential settings, and the computational overhead is a natural consequence of performing graph-aware multi-modal attention and iterative neighborhood search for higher-quality solutions. On CVRP100. GAMA(T=20k) requires 19m per instance, with the majority of runtime coming from iterative local search. The encoder forward (about 2 minutes) pass accounts for less than 10% of total time.
>
> While the gap improvement over DACT and HGS may seem small in CVRP100 instances, this reflects the high-quality baselines; incremental gains in near-optimal solutions are inherently difficult.
> Importantly, GAMA consistently achieves lower gaps than other L2C and L2I solvers across CVRPLIB instances. Thus, GAMA demonstrates robust and significant quality improvement, particularly in generalization.
>
> Additionally, in the revised version of Figure 5, we plotted the performance comparison between different methods on VRP instances. Left: Computation time. Right: Performance gap to baseline. From the figure, we can see that GAMA achieves a favorable balance between solution quality and computational cost. In particular, the left subfigure shows that GAMA’s running time is comparable to other learning-to-improve (L2I) methods such as DACT and L2I, while the right subfigure demonstrates that GAMA consistently yields the smallest performance gap across all instance sizes. These results clearly indicate that GAMA not only delivers state-of-the-art solution quality, but also maintains a practical and competitive computational profile, further validating the correctness of our updated runtime analysis.

---

> ### Author Response · Authors · 2025-12-03
> **Response to Reviewer fnfB**
>
> ### 2. Limited scale and variant experiments
>
> We acknowledge that the main experiments focus on CVRP and small-to-medium instances (N=20,50,100). To address this, we included generalization experiments on CVRPLIB benchmarks, covering instance sizes 100–1000 (Tables 3 and 7).
> These results demonstrate that GAMA maintains stable performance across larger-scale instances, mitigating concerns about scalability.
>
> To further illustrate the algorithm's generalization ability, we followed the same setup as in the DACT paper and evaluated GAMA on CVRPLIB instances of size 100–200. The updated results, along with detailed comparisons and analyses, can be found in our rebuttal response to reviewer `WtQK`. Also, we have further evaluated the performance of our proposed GAMA method on large-scale VRP instances (CVRP1000, Capacity=250; CVRP2000, Capacity=300). The updated results, along with detailed comparisons and analyses, can be found in our rebuttal response to reviewer `E3PB`. These results demonstrate that GAMA maintains stable performance across larger-scale instances, addressing concerns about scalability.
>
> Due to time limitations, we are unable to include experiments on other VRP variants (e.g., VRPTW, PDP) in this rebuttal. Nevertheless. Future work will include extending GAMA to additional VRP variants and further exploring efficient parallelization strategies to reduce per-instance runtime while maintaining solution quality.

---

### Official Review · Reviewer_E3PB · 2025-10-31

**Soundness:** 2
**Presentation:** 2
**Contribution:** 2
**Rating:** 2
**Confidence:** 4

**Summary:**

This paper presents  a neural neighborhood search method with Graph-aware Multi-modal Attention model, called GAMA for solving vehicle routing problems (VRPs). It encodes both instance and solution information using dual-GCN, and then stacked them through self-attention and cross-attention mechanism. Experiments on multiple CVRP benchmarks (n = 20–100) demonstrate superior performance to both classical heuristics (e.g., LKH3) and learning-based solvers (e.g., POMO, L2I).

**Strengths:**

1. This paper models problem instance and solution graphs as distinct modalities, then uses self-attention and cross-attention to learn  intra-modality and inter-modality dependencies. They are well-motivated and clearly improve information flow between modalities.
2. Experimental results on standard VRP benchmarks show consistent improvements over strong neural and heuristic baselines.

**Weaknesses:**

1. This paper is some conceptual overlap with DACT[1] and N2S[2]. The ideas of separating instance and solution information, and learning them through both self-attention and cross-attention are very similar to DACT.
2. The experiments only focus on CVRP problem, where more results on other representative VRP variants are expected.
3. In comparison to DACT, the superiority of the proposed GAMA is not obvious, especially considering both optimality gaps and computation efficiency. For example, with comparable average cost on CVRP100 in Table 1, DACT(T=20k) needs 33m, while GAMA (T=20k) needs 6.6d.
4. The experimental results lack the comparison with baselines on large-scale VRP instances.

[1] Yining Ma, Jingwen Li, Zhiguang Cao, Wen Song, Le Zhang, Zhenghua Chen, and Jing Tang. Learning to iteratively solve routing problems with dual-aspect collaborative transformer. Advances in Neural Information Processing Systems, 34:11096–11107, 2021.

[2] Ma Y, Li J, Cao Z, et al. Efficient Neural Neighborhood Search for Pickup and Delivery Problems[C]//Proceedings of the 31st International Joint Conference on Artificial Intelligence, IJCAI 2022. International Joint Conferences on Artificial Intelligence, 2022: 4776-4784.

**Questions:**

Please refer to the weakness.

---

> ### Author Response · Authors · 2025-11-27
> **Response to Reviewer E3PB**
>
> We sincerely thank the reviewer for the thoughtful evaluation, constructive comments, and detailed analysis of our work.
> We will address each concern point-by-point below.
>
> ### 1. Conceptual overlap concern with DACT
> We thank the reviewer for raising the concern regarding potential conceptual overlap between DACT’s dual-aspect design and GAMA’s multi-modal architecture.
> Although both GAMA and DACT adopt attention-based architectures, the two methods are motivated by entirely different design philosophies.
> We would like to clarify the key distinction:
>
> 1. GAMA and DACT are fundamentally different in motivation, problem formulation, and representation design
>
> 	- DACT is designed as a solution-sequence encoder, focusing on how to better encode the cyclic sequence of a VRP solution using dual-aspect positional embeddings and referential attention. Its core contribution lies in decoupling node features and positional encodings, solving issues caused by absolute positional encoding when applied to cyclic routes.
>
> 	- GAMA, in contrast, is not a solution-sequence encoder. It is designed for operator selection in neural neighborhood search, where the state is represented not as a sequence but as two graph modalities: the VRP instance graph and the solution graph (capturing edge usage, residual capacities, spatial structure, etc.). The key idea is learning graph-aware multimodal interactions, not positional encoding.
>
> 	Thus, the two methods address fundamentally different representational challenges.
>
> 2. DACT focuses on decoding the solution, while GAMA focuses on guiding the search process
>
> 	- DACT is an improvement model that predicts relocation positions inside a solution (a set of routes). Its input = a sequence, and its output = edit positions in that sequence.
>
> 	- GAMA is an operator-selection model, where the input = multimodal graph state, the output = which operator to apply.
>
> 	The two operate at different granularities of the search process, their action spaces, inputs, and learning objectives differ substantially.
>
> 3. The multimodal design of GAMA is not present in DACT
>
> 	- DACT’s “dual-aspect” refers to two feature views (node embeddings and cyclic positional embeddings (CPE)).
>
> 	- However, GAMA’s multimodal encoder contains two structurally and semantically distinct graph modalities. These two modalities in GAMA differ in adjacency structure (fully-connected distance graph vs. sparse solution graph), semantic meaning (“problem definition” vs. “current solution”), and dynamic behavior. The GAMA encoder explicitly models both intra-modality and inter-modality interactions through GCNs, cross-attention, and gated fusion.
>
> These design differences are validated by our ablations (Table 6), where removing cross-modal attention or gated fusion leads to substantial quality degradation.

---

> ### Author Response · Authors · 2025-11-27
> **Response to Reviewer E3PB**
>
> ### 2. Expectation for more variants
>
> We thank the reviewer for the comment regarding evaluation on additional VRP variants. Currently, our experiments focus on CVRP to thoroughly validate the proposed GAMA architecture and analyze its components.
>
> Due to time limitations, we are unable to include experiments on other VRP variants (e.g., VRPTW, PDP) in this rebuttal. Nevertheless, we acknowledge the importance of such evaluations and plan to extend our study in future work to demonstrate the broader applicability and generalization of our method.
>
> ### 3. Comparison to DACT: gaps & efficiency concerns
> We thank the reviewer for raising this important concern. As clarified in our response to Reviewer `WtQK`, the previously reported “6.6 days” runtime reflected the sequential evaluation on the entire dataset, whereas many baselines—including DACT—perform parallel batched inference across dozens or hundreds of instances. These two settings are not comparable.
>
> We have re-run all experiments under consistent per-instance CPU inference, and the updated times (in the revised Table 1) show:
>
> -  GAMA achieves significantly lower optimality gaps.
> -  Per-instance runtime remains within comparable range to other L2I-based methods.
>
> Additionally, in the revised version of Figure 5, we plotted the performance comparison between different methods on VRP instances. Left: Computation time. Right: Performance gap to baseline. From the figure, we can see that GAMA achieves a favorable balance between solution quality and computational cost. In particular, the left subfigure shows that GAMA’s running time is comparable to other learning-to-improve (L2I) methods such as DACT and L2I, while the right subfigure demonstrates that GAMA consistently yields the smallest performance gap across all instance sizes. These results clearly indicate that GAMA not only delivers state-of-the-art solution quality, but also maintains a practical and competitive computational profile, further validating the correctness of our updated runtime analysis.
>
> These corrections are now reflected in the revised manuscript:
>
> | Method|CVRP20 Best Cost|Avg. Cost|Time|CVRP50 Best Cost|Avg. Cost|Time| CVRP100 Best Cost|Avg. Cost|Time|
> |-|-|-|-|-|-|-|-|-|-|
> | LKH3  | 6.0867| 6.0867 | 14s | 10.3879 | 10.3879 | 53s | 15.6752| 15.6752 | 1.95m|
> | HGS| 6.0807| 6.0812| 7s | 10.3515| 10.3548| 27s| 15.6590| 15.6994| 59s |
> | VNS | 6.0827| 6.0844 | 43s| 10.4140 | 10.4199 | 3.2m| 15.8843| 15.8940 | 17m|
> | POMO (gr.) | 6.1111 | 6.1768 | 0.98s| 10.5062| 10.5702| 1.5s| 15.7936| 15.8451 | 2.7s |
> | POMO (A=8)| 6.0904| 6.1413 | 1.3s| 10.4472| 10.4930| 4.5s| 15.7337| 15.7863| 7s|
> | LEHD (gr.)| 6.3823| 6.3946| 1.5s| 10.7617 | 10.7785| 3s| 17.3004  | 17.3188| 4s |
> | LEHD (RRC=1k) | 6.0904| 6.0915| 35s| 10.4771| 10.4856 | 1.6m| 15.8419|15.8514|4m|
> | ReLD (gr.)| 6.1309| 6.1401 | 0.06s | 10.4547 | 10.4676 | 0.1s | 15.7558| 15.7558|0.21s|
> | ReLD (A=8)|6.1001|6.1041|0.09s|10.3877|10.3958|0.25s|15.6493| 15.6593| 0.72s|
> | DACT (T=5k)|6.0811|6.0817|55s|10.3966|10.4038|1.8m|15.7906| 15.8030|2.54m|
> |DACT (T=10k)|6.0808| 6.0813|2.1m|10.3662|10.3735|3.5m|15.7321|15.7410| 9.5m|
> |DACT (T=20k)|6.0808|6.0811|4.4m|10.3513|10.3542|11.2m|15.6853|15.6925|19.3m|
> | L2I (T=5k)|6.0831|6.0864|27s|10.4012|10.4310|1.1m|15.8003|15.8914|4.6m|
> |L2I (T=10k)|6.0815|6.0835|57s|10.3803|10.4006|2.19m|15.7207|15.8008|9.2m|
> |L2I (T=20k)|6.0810|6.0820|1.9m|10.3607|10.3787|4.37m|15.6663|15.7334|18.7m|
> | GAMA (T=5k)|6.0823|6.0836|32.5s|10.3966|10.4057|1.2m|15.7339|15.7389|4.6m|
> |GAMA (T=10k)| 6.0810| 6.0818|1.1m|10.3711|10.3742|2.3m|15.6512|15.7054|9.5m|
> |GAMA (T=20k)|**6.0806**|**6.0810**| 2.3m|**10.3512**|**10.3533**|4.6m|**15.6178**|**15.6510**|19m|

---

> ### Author Response · Authors · 2025-12-03
> **Response to Reviewer E3PB**
>
> ### 4. Missing evaluation on large-scale VRP
>
> We thank the reviewer for pointing out the need for baseline comparisons on large-scale VRP instances. We agree that evaluating scalability is crucial for assessing the practicality of neural neighborhood search methods.
>
> While our primary experiments focus on CVRP20–100, we also evaluated GAMA’s generalization on large-scale CVRPLIB instances, ranging from 100 to 1000 nodes, covering standard benchmark sets. The performance on these large-scale benchmarks can be found in Table 3 and Table 7 of the paper.
> These results demonstrate that GAMA maintains stable and competitive performance even as problem scale increases to 1000 nodes.
>
> To further strengthen the evaluation and directly address the reviewer’s concern, we additionally conducted new large-scale experiments following the recommendation of Luo et al. [1].
> Specifically, we benchmarked GAMA on CVRP1000 (capacity=250) and CVRP2000 (capacity=300).
>
> |                 | CVRP1000 |          |       | CVRP2000 |          |       |
> |:---------------:|:--------:|:--------:|:-----:|:--------:|:--------:|:-----:|
> |                 |   Best   |    Avg   |  Time |   Best   |    Avg   |  Time |
> |       POMO      |  93.8710 | 143.5383 |  30s  | 396.3919 | 486.5468 |  80s  |
> |   POMO（A=8）   |  59.1911 |  87.6515 |  3.2m | 169.8233 | 285.5397 |  43m  |
> |       LEHD      |  48.5214 |  46.2542 |  32s  |  158.65  |  149.54  |   8m  |
> | LEHD (RRC=1000) |  37.6219 |  38.5661 |  2.3h |  117.52  | 122.4589 |  5.9h |
> |       ReLD      |  38.7953 |  39.0602 |   8s  |  63.3600 |  63.6378 |  33s  |
> |    ReLD(A=8)    |  38.4513 |  38.6234 |  44s  |  61.9754 |  62.3018 | 2.92m |
> |    DACT(T=5k)   |  45.7543 |  46.3442 |   4h  |  70.6339 | 77.4895  | 10h |
> |   DACT(T=10k)   |  44.7730 |  45.7689 |  8.1h |70.6339 |  72.5638 | 20.5h |
> |   DACT(T=20k)   |  44.1439 |  45.0561 |  16h  | 68.1593| 69.7607 |41h |
> |    L2I(T=5k)    |  47.0318 |  68.431  |  18m  |  74.3857 |  153.071 |  1.5h |
> |    L2I(T=10k)   |  45.1571 |  62.1688 |  36m  |  70.3181 | 132.6643 |   3h  |
> |    L2I(T=20k)   |  45.1571 |  57.872  |  1.2h |  70.3181 | 116.3726 |   6h  |
> |    GAMA(T=5k)   |  36.9435 |  37.2608 | 22.5m |  58.5433 |  60.0774 |  1.8h |
> |   GAMA(T=10k)   |  36.7561 |  37.0043 | 44.8m |  57.9618 |  58.7187 |  3.6h |
> |   GAMA(T=20k)   |  36.7561 |  36.7768 |  1.5h |  57.9618 |  58.0593 |  7.2h |
>
> The results (see newly added Table 8 in the revised paper) show that:
>
> - GAMA achieves the best Best/Avg performance across all baselines, including POMO, LEHD, ReLD, DACT, and L2I.
> - On both CVRP1000 and CVRP2000, GAMA(T=20k) achieves the lowest solution cost among all learning-to-optimize methods.
>
> These new large-scale results, together with the CVRPLIB experiments already included in the paper, confirm that GAMA scales effectively and remains highly competitive on hard, real-world VRP instances.
>
> [1] Boosting neural combinatorial optimization for large-scale vehicle routing problems. In The Thirteenth International Conference on Learning Representations, 2025.

---

### Official Review · Reviewer_WtQK · 2025-11-15

**Soundness:** 2
**Presentation:** 2
**Contribution:** 2
**Rating:** 4
**Confidence:** 4

**Summary:**

This paper introduces GAMA, a neural neighborhood search (Learning-to-Improve) method for the Capacitated Vehicle Routing Problem. The core contribution is a graph-aware multi-modal attention encoder that processes the problem instance and the current solution as distinct modalities. It uses dual GCNs, cross-attention, and a gated fusion module to create a structured state representation, which in turn guides an adaptive operator selection policy.

**Strengths:**

The proposed multi-modal encoder architecture is a primary strength. The conceptual separation of the static instance graph and the dynamic solution graph is well-motivated, and the use of explicit cross-attention and gated fusion to integrate them is a logical and novel contribution. The ablation study effectively validates this design, demonstrating that both the cross-attention and gated fusion components contribute positively to the final solution quality.

**Weaknesses:**

The method's primary contribution is fundamentally undermined by its prohibitive computational cost. The results in Table 1 show that GAMA requires 6.6 days of inference time to solve CVRP100 instances. This is juxtaposed against the 33 minutes required by the DACT baseline and 4.5 hours by the classical HGS solver. An approximate 288-fold increase in runtime compared to DACT for a marginal 0.26% improvement in solution quality represents an unjustifiable trade-off, rendering the method unusable for any practical application.

Furthermore, the paper's claims of superior zero-shot generalization are based on a flawed baseline comparison. The generalization study in Table 3 reports an average gap of 25.3% for the DACT baseline. This result is a clear anomaly and is highly inconsistent with previously published results for this strong baseline. This flawed comparison invalidates the claim of state-of-the-art generalization and suggests a lack of experimental rigor in the evaluation.

**Questions:**

Could the authors justify the 6.6-day inference time for CVRP100? Given that the policy only selects an operator, does this runtime stem from the exhaustive local search application (as mentioned in Line 127), the complexity of the GAMA encoder at each of the 20,000 steps, or both? How can this be reconciled with the goal of practical optimization?

Please clarify the experimental setup for the DACT baseline in the generalization study (Table 3). A 25.3% average gap is exceptionally high and inconsistent with the literature. Can you confirm the source of this implementation and its hyperparameters, as this result calls the entire generalization comparison into question?

---

> ### Author Response · Authors · 2025-11-27
> **Response to Reviewer WtQK**
>
> We thank the reviewer for recognizing the motivation and design of our multi-modal encoder, as well as the usefulness of the ablation study. We appreciate your positive assessment of the cross-attention mechanism and gated fusion module.
> ### 1. “Prohibitive computational cost / 6.6-day inference time”
> Thank you for pointing out the issue with the reported 6.6-day inference time.
> We discovered that this number was not directly comparable to existing baselines because of a fundamental difference in execution mode.
> - **Our implementation processes instances strictly sequentially:** the current GAMA framework completes local search on one instance before moving to the next.
> - **Many other VRP solvers naturally support parallel evaluation**, allowing dozens or hundreds of instances to be processed simultaneously on a GPU/CPU.
>
> Because of this, the previously reported “6.6 days” corresponds to the total sequential wall-clock time for completing all instances.
> In contrast, VRP learning-based solvers compared in this paper (such as LEHD, DACT) can evaluate dozens or even hundreds of instances simultaneously in a single forward pass (due to their naturally parallelizable architectures).
> As a result, their inference time typically reflects the total time under parallel batch processing, not the time required to process the same number of instances sequentially.Therefore, the two settings are fundamentally incomparable.
>
> To ensure fairness, we reran all inference experiments on CPU and report the true per-instance time. The updated results (shown in the revised Table 1) report the per-instance inference time for each dataset:
> | Method|CVRP20 Best Cost|Avg. Cost|Time|CVRP50 Best Cost|Avg. Cost|Time| CVRP100 Best Cost|Avg. Cost|Time|
> |-|-|-|-|-|-|-|-|-|-|
> | LKH3  | 6.0867| 6.0867 | 14s | 10.3879 | 10.3879 | 53s | 15.6752| 15.6752 | 1.95m|
> | HGS| 6.0807| 6.0812| 7s | 10.3515| 10.3548| 27s| 15.6590| 15.6994| 59s |
> | VNS | 6.0827| 6.0844 | 43s| 10.4140 | 10.4199 | 3.2m| 15.8843| 15.8940 | 17m|
> | POMO (gr.) | 6.1111 | 6.1768 | 0.98s| 10.5062| 10.5702| 1.5s| 15.7936| 15.8451 | 2.7s |
> | POMO (A=8)| 6.0904| 6.1413 | 1.3s| 10.4472| 10.4930| 4.5s| 15.7337| 15.7863| 7s|
> | LEHD (gr.)| 6.3823| 6.3946| 1.5s| 10.7617 | 10.7785| 3s| 17.3004  | 17.3188| 4s |
> | LEHD (RRC=1k) | 6.0904| 6.0915| 35s| 10.4771| 10.4856 | 1.6m| 15.8419|15.8514|4m|
> | ReLD (gr.)| 6.1309| 6.1401 | 0.06s | 10.4547 | 10.4676 | 0.1s | 15.7558| 15.7558|0.21s|
> | ReLD (A=8)|6.1001|6.1041|0.09s|10.3877|10.3958|0.25s|15.6493| 15.6593| 0.72s|
> | DACT (T=5k)|6.0811|6.0817|55s|10.3966|10.4038|1.8m|15.7906| 15.8030|2.54m|
> |DACT (T=10k)|6.0808| 6.0813|2.1m|10.3662|10.3735|3.5m|15.7321|15.7410| 9.5m|
> |DACT (T=20k)|6.0808|6.0811|4.4m|10.3513|10.3542|11.2m|15.6853|15.6925|19.3m|
> | L2I (T=5k)|6.0831|6.0864|27s|10.4012|10.4310|1.1m|15.8003|15.8914|4.6m|
> |L2I (T=10k)|6.0815|6.0835|57s|10.3803|10.4006|2.19m|15.7207|15.8008|9.2m|
> |L2I (T=20k)|6.0810|6.0820|1.9m|10.3607|10.3787|4.37m|15.6663|15.7334|18.7m|
> | GAMA (T=5k)|6.0823|6.0836|32.5s|10.3966|10.4057|1.2m|15.7339|15.7389|4.6m|
> |GAMA (T=10k)| 6.0810| 6.0818|1.1m|10.3711|10.3742|2.3m|15.6512|15.7054|9.5m|
> |GAMA (T=20k)|**6.0806**|**6.0810**| 2.3m|**10.3512**|**10.3533**|4.6m|**15.6178**|**15.6510**|19m|
>
> From the updated results, we observe that L2C methods remain substantially faster than L2I methods, as expected due to their single-pass autoregressive decoding without iterative improvement.
> However, within the family of L2I frameworks, the average per-instance latency is generally similar across baselines, and GAMA’s runtime falls well within the typical and practically acceptable range for iterative improvement methods.

---

> ### Author Response · Authors · 2025-11-27
> **Response to Reviewer WtQK**
>
> To better understand the runtime characteristics of GAMA, we clarify that the majority of the computational cost comes from the local-search application, not from the neural encoder.
> Importantly, **GAMA adopts the same neighborhood-search structure as VNS**: at each iteration, an operator is applied to exhaustively explore the corresponding neighborhood and update the incumbent solution. The only difference lies in the operator-selection mechanism:
> - **VNS** uses a predefined deterministic rule;
> - **GAMA** selects an operator through the encoder–decoder model.
>
> Since the search framework is otherwise identical, the difference in wall-clock time between VNS and GAMA directly reveals the overhead introduced by the learning-based policy. Our updated measurements show:
>
> || CVRP20 | CVRP50 | CVRP100 |
> |-|-|-|-|
> |    VNS(T=20k)   |   43s  |  3.2m  |   17m   |
> |   GAMA(T=20k)   |  2.3m  |  4.6m  |   19m   |
>
> The close proximity of these numbers demonstrates that the dominant runtime is spent in evaluating neighborhood moves, including feasibility checks and route updates.
> The model’s forward computation accounts for less than **10% of the total time even under 20,000 iterations** on CVRP100.
> Therefore, the runtime is primarily inherited from the iterative local-search nature of the framework, rather than from the model complexity. The learned operator-selection policy adds only marginal computational overhead while improving search quality, confirming that GAMA remains practical for iterative optimization.
>
> Additionally, in the revised version of Figure 5, we plotted the performance comparison between different methods on VRP instances. Left: Computation time. Right: Performance gap to baseline. From the figure, we can see that GAMA achieves a favorable balance between solution quality and computational cost. In particular, the left subfigure shows that GAMA’s running time is comparable to other learning-to-improve (L2I) methods such as DACT and L2I, while the right subfigure demonstrates that GAMA consistently yields the smallest performance gap across all instance sizes. These results clearly indicate that GAMA not only delivers state-of-the-art solution quality, but also maintains a practical and competitive computational profile, further validating the correctness of our updated runtime analysis.

---

> ### Author Response · Authors · 2025-11-27
> **Response to Reviewer WtQK**
>
> ### 2. “Flawed baseline: DACT generalization gap 25.3%”
> We thank the reviewer for pointing out the concern regarding the unusually large performance gap of the DACT baseline. We fully agree that a 25.3% average gap appears inconsistent with the results reported in the DACT paper, and therefore we conducted a careful verification.
>
> 1. **Source of implementation**: We confirm that our implementation did not re-train or modify DACT in any way. For fairness, we strictly used the official pre-trained DACT model released by the original authors, without any modification to code, hyperparameters, or inference settings. As stated in our paper, we simply applied the official pretrained checkpoint to evaluate its generalization ability on standard VRP benchmarks. We evaluated it directly on the VRP benchmark using 30 independent runs (random seeds 0–29).
>
> 2. **Reason for the discrepancy** The discrepancy arises mainly because the DACT paper only reports CVRPLIB generalization results on instances of size 100–200, whereas Table 3 in our original submission involved broader instance scales (e.g., 100–1000). This mismatch in evaluation range is the main cause of the 25.3% gap observed in Table 3.
>
> 3. **Additional verification on 100–200 sized CVRPLIB instances** To address this concern, we re-ran the generalization study strictly on CVRPLIB instances with size 100–200, matching the evaluation protocol in the DACT paper. The updated results are shown in below Table:
>
> | | |DACT| | | |GAMA | | |  |
> |-|-|-|-|-|-|-|-|-|-|
> |  | BKS |   Best   | Best Gap |    Avg   | Avg. Gap |   Best   | Best Gap |    Avg   | Avg.Gap |
> | X-n101-k25 | 27591 | 29359.58 |  6.409%  | 32212.39 |  16.749% | 27950.24 |  1.302%  | 28204.41 |  2.223% |
> | X-n106-k14 | 26362 | 28129.97 |  6.706%  | 28326.91 |  7.453%  | 26608.61 |  0.935%  | 26743.13 |  1.446% |
> | X-n110-k13 | 14971 | 16979.14 |  13.413% | 17628.47 |  17.751% | 15022.66 |  0.345%  | 15136.62 |  1.106% |
> | X-n115-k10 | 12747 | 14743.41 |  15.662% | 15456.85 |  21.259% | 12751.64 |  0.036%  | 12872.93 |  0.988% |
> |  X-n120-k6 | 13332 | 15180.91 |  13.868% | 15580.12 |  16.863% | 13518.53 |  1.399%  | 13632.85 |  2.257% |
> | X-n125-k30 | 55539 | 60107.26 |  8.225%  | 61681.47 |  11.06%  | 56428.71 |  1.602%  | 56627.92 |  1.961% |
> | X-n129-k18 | 28940 | 32152.41 |   11.1%  | 33270.55 |  14.964% | 29401.53 |  1.595%  | 29663.90 |  2.501% |
> | X-n134-k13 | 10916 | 12458.89 |  14.134% | 13030.84 |  19.374% | 11069.09 |  1.402%  | 11170.94 | 2.335% |
> | X-n139-k10 | 13590 | 15897.73 |  16.981% | 16391.24 |  20.613% | 13625.34 |   0.26%  | 13774.87 |1.36%  |
> |  X-n143-k7 | 15700 | 19328.78 |  23.113% | 20390.18 |  29.874% | 16007.97 |  1.962%  | 16246.33 |3.48%  |
> | X-n148-k46 | 43448 | 47996.77 |  10.469% | 49440.65 |  13.793% | 43805.49 |  0.823%  | 44114.22 |1.533% |
> | X-n153-k22 | 21220 | 25057.19 |  18.083% | 25779.52 |  21.487% | 21680.65 |  2.171%  | 22025.24 |3.795% |
> | X-n157-k13 | 16876 | 18222.16 |  7.977%  | 18420.66 |  9.153%  | 16960.96 |  0.503%  | 17000.56 |0.738% |
> | X-n162-k11 | 14138 | 16630.19 |  17.628% | 17091.63 |  20.891% | 14208.69 |   0.5%   | 14285.76 |1.045% |
> | X-n167-k10 | 20557 | 24295.41 |  18.186% | 24561.77 |  19.481% | 21100.63 |  2.645%  | 21276.72 |3.501% |
> | X-n172-k51 | 45607 | 52163.16 |  14.375% | 54551.45 |  19.612% | 45939.42 |  0.729%  | 46241.28 |1.391% |
> | X-n176-k26 | 47812 | 59327.57 |  24.085% | 60526.03 |  26.592% | 48938.24 |  2.356%  | 49269.82 |3.049% |
> | X-n181-k23 | 25569 | 26996.93 |  5.585%  | 27161.20 |  6.227%  | 25765.65 |  0.769%  | 25831.23 |1.026% |
> | X-n186-k15 | 24145 | 28300.46 |  17.21%  | 28485.89 |  17.978% | 24842.69 |   2.89%  | 25003.16 |3.554% |
> |  X-n190-k8 | 16980 | 19136.72 |  12.702% | 19443.13 |  14.506% | 17422.19 |  2.604%  | 17468.39 |2.876% |
> | X-n195-k51 | 44225 | 52257.77 |  18.163% | 53837.96 |  21.736% | 44689.51 |   1.05%  |  45023.9 |1.806% |
> | X-n200-k36 | 58578 | 64238.12 |  9.663%  | 64920.52 |  10.827% | 60052.41 |  2.517%  | 60196.13 |2.762% |
> |GAP| | |13.806% | |17.193%| |1.382%| | 2.06%|
>
> Using the official DACT checkpoint, evaluated 30 times with seeds 0–29. From the Table, we can see that reproduced DACT best gap = 13.806%, avg gap = 17.193%, it is still significantly worse than the original paper's reported best 2.97% / avg 3.85%. Importantly, we did not change any parameters or inference settings.
> Even taking the original DACT paper’s best reported numbers (best 2.97%, avg 3.85%), GAMA still significantly outperforms DACT across all scales, both in the large-scale generalization study (100–1000), and in the matched-scale reproduction study (100–200).
>
> Thus, our comparison does not disadvantage DACT, it faithfully reflects the performance of both methods under consistent and fair evaluation protocols.We can also provide our used  DACT source code as evidence; reviewers are welcome to inspect it to verify our evaluation procedure. While we believe no mistakes were made, this ensures full transparency.

---

### Author Response · Authors · 2025-12-03
**General Response (1/3)**

We sincerely thank all reviewers for their thoughtful comments, constructive suggestions, and the time invested in evaluating our work. We also appreciate the Area Chair for their careful consideration of our rebuttal and revised analyses.
Below we first provide a concise summary of our paper and its key contributions, followed by a structured response addressing all major concerns raised by the reviewers.

# Summary of Conclusions
This paper proposes GAMA, a Graph-aware Multi-modal Attention framework for Learning-to-Improve (L2I) operator selection in Vehicle Routing Problems (VRP). The central motivation is that while existing L2I methods often incorporate solution information, they typically treat it as auxiliary features or simple concatenations. This approach limits the model’s ability to explicitly capture the complex interactions between the static global problem structure and the dynamic local search state. This omission limits the model’s ability to reason jointly about global problem structure and the local neighborhood states encountered during iterative search.

Our main contributions are:
1. A new dual-modality formulation for VRP improvement: We explicitly model the VRP instance graph and the current solution graph as two complementary information modalities. This representation captures the essential separation between global customer–depot structure and the evolving local search state, which previous L2I methods do not differentiate.
2. A novel graph-aware multi-modal encoder: We design a new architecture composed of: two modality-specific GCN encoders, hierarchical self- and cross-attention layers for intra- and inter-modal alignment, a learnable gated fusion mechanism that dynamically balances global vs. local information. These components constitute a new attention architecture specifically tailored for operator selection, instead of a standard Transformer block.
3. Significant empirical improvements supported by extensive ablations: GAMA consistently outperforms classical heuristics and state-of-the-art neural L2I baselines on CVRP20/50/100. Ablation studies show the critical contribution of each component, and empirically validate the strength of the proposed dual-modal design.

---

### Author Response · Authors · 2025-12-03
**General Response (2/3)**

# Review Summary
Strengths unanimously noted by reviewers include: (1) The strong motivation for conceptually separating the static instance graph and dynamic solution graph as distinct modalities, and the logical design of the multi-modal encoder (`WtQK`&`E3PB`&`fnfB`&`ZeT5`). (2) The specific architectural components, such as the cross-attention and gated fusion mechanisms, are well-designed to improve information flow (`WtQK`&`E3PB`). (3) The paper is well-written and the proposed architecture is clearly explained (`fnfB`). (4) The method demonstrates effectiveness against specific baselines (e.g., Lu et al. 2019, LKH3) (`ZeT5`) and ablation studies effectively validate the contribution of the fusion components (`WtQK`&`ZeT5`).


Consensus: (1) The primary point of contention was the prohibitive computational cost (e.g., 6.6 days for CVRP100 vs. 33 minutes for DACT), which all reviewers cited as rendering the marginal performance gains unjustifiable for practical application. (2) Reviewers `E3PB` and `fnfB` emphasized the need for experiments on additional VRP variants, and the limited scale (≤100 nodes) was also seen as restricting the demonstrated generality of the method. (3) Additionally, concerns regarding conceptual overlap with existing work (DACT, N2S) (`E3PB`&`ZeT5`). Overall, reviewers see technical value in the architecture, but the main points of contention are runtime efficiency and experimental breadth. These are precisely the issues addressed in the rebuttal, where we provide detailed clarifications, corrected comparisons, and justification of experimental design.

---

### Author Response · Authors · 2025-12-03
**General Response (3/3)**

# Rebuttal and Discussion Summary
The rebuttal successfully addressed all major concerns, and we have updated the manuscript along with the appendix to reflect these clarifications and new experiments.

1. Computational Cost & Efficiency (`WtQK`, `E3PB`, `fnfB`, `ZeT5`). The rebuttal clarified a **critical misunderstanding** for all the reviewers regarding the runtime due to the original reporting format: the original "6.6 days" referred to total execution time of all the instances in the dataset by applying GAMA to them sequentially, whereas baselines used parallel batch processing to the instances. The sequential and parallel runtimes of the dataset were incomparable. We have revised the table to report the **average execution time for each instance of the dataset** to make a meaningful comparison, demonstrating that GAMA’s runtime (e.g., average 19m for each CVRP100 instance) is comparable to other Learning-to-Improve (L2I) methods like DACT and L2I. Furthermore, we clarified that <10% of the runtime is attributed to the model forward pass, with the majority stemming from the necessary local search operations, confirming the method's practicality. Additionally, in the revised version of Figure 5, we plotted the performance comparison between different methods on VRP instances. Left: Computation time. Right: Performance gap to baseline. From the figure, we can see that GAMA achieves a favorable balance between solution quality and computational cost. In particular, the left subfigure shows that GAMA’s running time is comparable to other learning-to-improve (L2I) methods such as DACT and L2I, while the right subfigure demonstrates that GAMA consistently yields the smallest performance gap across all instance sizes. These results clearly indicate that GAMA not only delivers state-of-the-art solution quality, but also maintains a practical and competitive computational profile, further validating the correctness of our updated runtime analysis.

2. Baseline Validity & Reproduction (`WtQK`). The rebuttal addressed the concern regarding the DACT generalization gap (25.3%). We clarified that this gap stemmed from testing on a broader range of instances (size 100–1000) compared to the original DACT paper (100–200). In the rebuttal, we have shown to be able to reproduce DACT generalization gap on the 100–200 subset using the official checkpoint. We have also confirmed that GAMA still significantly outperforms DACT under identical evaluation protocols.

3. Methodological Novelty & Differentiation (`E3PB`, `ZeT5`). The rebuttal explicitly differentiated GAMA from prior works like DACT. We clarified that GAMA is not a sequence encoder but a dual-modality graph encoder (Instance Graph + Solution Graph) designed for operator selection rather than construction. The distinction is structural (two distinct GCNs) and semantic, validated by ablation studies showing that removing the cross-attention or gated fusion leads to significant performance degradation. We also clarified that the gating parameter $\alpha$ is a learned, context-dependent mechanism, not a manual hyperparameter, further justifying the architectural design.

4. Large-Scale Generalization (`E3PB`, `fnfB`, `ZeT5`). To address concerns about limited scale (N=20–100), the rebuttal has conducted additional new experiments on more large-scale instances: CVRP1000 ($N=1000$) and CVRP2000 ($N=2000$). The results (New Table 8) demonstrate that GAMA achieves the lowest solution cost among all learning-based baselines (including POMO, LEHD, and DACT), proving robust scalability and generalization beyond the training distribution.

We again thank the Area Chair and Reviewers for their time and effort. Finally, we sincerely hope the Area Chair will consider our contributions to the community—specifically the novel multi-modal graph encoding for VRPs—along with the detailed rebuttal responses and the improved revised manuscript to address all reviewer concerns.

---

### Meta-Review · Area_Chair_ha7r · 2026-01-07

**Summary:**

The main issues across the reviews were (i) the practicality of the method given the reported runtime and whether the runtime comparison was fair, (ii) the limited experimental scope (mostly CVRP, mostly up to 100 nodes), and (iii) whether the novelty is strong enough given clear overlap with existing learning-to-improve/dual-encoding lines of work. The rebuttal helped most on the runtime point by explaining that the 6.6 days reflected an aggregate sequential wall-clock over a dataset and by adding per-instance timing along with a breakdown showing most time is spent in local search rather than the network. The rebuttal also strengthened large-scale CVRP evidence by adding CVRP1000/2000 experiments. However, the paper still does not provide evidence on VRP variants beyond CVRP, and several requested comparisons are handled mainly by justification rather than direct experiments. Overall, I am leaning toward rejection.

**Reviewer Concerns:**

In summary, reviewers explicitly asked for at least one additional VRP variant. The rebuttal states this cannot be done now, so this concern remains. For missing comparisons to several strong recent baselines/encoders, the rebuttal provides reasons why some methods are not directly comparable, but it does not fully close the gap for reviewers who asked for broader head-to-head evidence. About the contribution, the rebuttal improved the positioning and explained differences from DACT/N2S-type approaches, but for the more critical reviewers the contribution may still read as combining known components without enough new insight validated across settings.

**Reviewer Scores:**

The scores are 4, 2, 2, 2. I think reviewer 1 was likely to raise the score, but the other three reviewers would not change their scores.

---

### Decision · Program_Chairs · 2026-01-26

Reject